

# The CHROMA cloud top pressure retrieval algorithm for the Plankton, Aerosol, Cloud, ocean Ecosytem (PACE) satellite mission

Andrew M. Sayer[1,2], Luca Lelli[3,4], Brian Cairns[5], Bastiaan van Diedenhoven[6], Amir Ibrahim[2], Kirk D. Knobelspiesse[2], Sergey Korkin[1,2], and P. Jeremy Werdell[2]

[1]Goddard Earth Sciences Technology And Research (GESTAR) II, University of Maryland Baltimore County, Baltimore, MD, USA
[2]NASA Goddard Space Flight Center, Greenbelt, MD, USA
[3]Institute of Remote Sensing Methods, German Aerospace Agency (DLR),Wessling, Germany
[4]Institute of Environmental Physics and Remote Sensing, University of Bremen, Bremen, Germany
[5]NASA Goddard Institute for Space Studies, New York, NY, USA
[6]SRON Netherlands Institute for Space Research, Leiden, Netherlands

**Correspondence:** Andrew M. Sayer (andrew.sayer@nasa.gov)

**Abstract.** This paper provides the theoretical basis and simulated retrievals for the Cloud Height Retrieval from $O_2$ Molecular Absorption (CHROMA) algorithm. Simulations are performed for the Ocean Color Instrument (OCI), which is the primary payload on the forthcoming NASA Plankton, Aerosol, Cloud, ocean Ecosystem (PACE) mission, and the Ocean Land Colour Instrument (OLCI) currently flying on the Sentinel 3 satellites. CHROMA is a Bayesian approach which simultaneously re-
trieves cloud optical thickness (COT), cloud top pressure/height (CTP/CTH), and (with a significant prior constraint) surface albedo. Simulated retrievals suggest that the sensor and algorithm should be able to meet the PACE mission goal for CTP error, which is ±60 mb for 65 % of opaque (COT≥ 3) single-layer clouds on global average. CHROMA will provide pixel-level uncertainty estimates, which are demonstrated to have skill at telling low-error situations from high-error ones. CTP uncertainty estimates are well-calibrated in magnitude, although COT uncertainty is overestimated relative to observed errors. OLCI
performance is found to be slightly better than OCI overall, demonstrating that it is a suitable proxy for the latter in advance of PACE's launch. CTP error is only weakly sensitive to correct cloud phase identification or assumed ice crystal habit/roughness. As with other similar algorithms, for simulated retrievals of multi-layer systems consisting of optically thin cirrus clouds above liquid clouds, retrieved height tends to be underestimated because the satellite signal is dominated by the optically-thicker lower layer. Total (liquid plus ice) COT also becomes underestimated in these situations. However, retrieved CTP becomes
closer to that of the upper ice layer for ice COT≈3 or higher.

## 1   Introduction

The NASA Plankton, Aerosol, Cloud, ocean Ecosystem (PACE) mission (https://pace.gsfc.nasa.gov) is expected to launch in January 2024. As its name suggests, it will extend and improve upon various key satellite-based Earth system data records in the atmospheric and ocean colour disciplines (Werdell et al., 2019). PACE will host three sensors; the primary payload
is the Ocean Color Instrument (OCI), a broad-swath passive imaging radiometer with continuous spectral coverage from the



ultraviolet to near-infrared (NIR), and seven discrete channels in the NIR and shortwave infrared (SWIR). It will also carry two cubesat-sized multi-angle polarimeters, namely the Hyper-Angle Rainbow Polarimeter 2 (HARP2; Martins et al., 2018) and Spectro-Polarimeter for planetary EXploration (SPEXone; van Amerongen et al., 2019).

Mission success requires routine production of various geophysical data sets from OCI measurements. These 'required' data

products represent a minimal set of those which it is envisioned the mission will be able to produce, and in most cases the at-launch algorithms are expected to be adaptations of existing heritage retrieval codes. Due to differences in sensor capabilities, some algorithms require more adaptation than others. For clouds, the required products include a cloud mask, cloud optical thickness (COT) at mid-visible wavelengths, cloud effective radius (CER), cloud top pressure (CTP), and cloud phase (i.e. liquid droplets or ice crystals). Together COT, CER, and phase will also be used to calculate cloud water path (CWP). CTP can

be transformed to cloud top height or temperature (CTH, CTT respectively); in this paper 'altitude' is used to refer to these three coordinate systems generically. These are the same cloud properties in operational production from NASA's MODerate resolution Imaging Spectroradiometer (MODIS) and Visible Infrared Imaging Radiometer Suite (VIIRS) sensors (Platnick et al., 2003), among others.

It is envisioned that the at-launch cloud algorithms will follow the same basic processing structure as the MODIS/VIIRS

heritage; that is, separate algorithms for cloud mask, COT/CER/CWP (often termed 'cloud optical properties'), phase, and CTP/CTH/CTT (Platnick et al., 2003, 2021). At present, cloud optical properties code equivalent to the operational Collection 6.1 MODIS product (Platnick et al., 2017) has been implemented into the PACE data processing stream, an algorithm to retrieve phase is in development (Coddington et al., 2017), and several cloud masking approaches are in consideration. This paper concerns the remaining gap: the cloud top altitude. MODIS, VIIRS, and several other sensors employ multispectral

thermal infrared (TIR) measurements, making use of thermal contrast between the (warm) surface and (colder) cloud tops, together with ancillary temperature-pressure-height profiles (e.g. Menzel et al., 2008; Heidinger and Pavolonis, 2009; Poulsen et al., 2012). PACE, however, will not measure at TIR wavelengths (see Section 2) so the MODIS-like approach cannot be adopted. Other techniques exist, such as hyperspectral TIR sounding, parallax, and lidar, radar, and microwave, as well as absorption features in the solar spectral region.

Early discussion on determining cloud top altitude from space using solar measurements suggested using channels in absorption lines for well-mixed gases. Note that satellite measurements are often referred to as 'channels' or 'bands' synonymously; in this paper, for clarity, 'channels' are used to discuss satellite measurements while 'bands' refers to atmospheric absorption regions. Then, altitude can be determined by the extent to which these absorption features are filled in by a scattering cloud compared to a nearby absorption-free window reference channel. Hanel (1961) initially suggested using $CO_2$ absorption near

$2\,\mu$m for this purpose, though Yamamoto and Wark (1961) and Chapman (1962) countered with the proposal of the $O_2$ A-band near 760 nm due to stronger cloud reflectance and stronger atmospheric absorption. Over the next several years theoretical and instrumental progress was made (e.g. Saiedy et al., 1965; Wark and Mercer, 1965), allowing airborne measurements (Saiedy et al., 1965) and then the first spaceborne observations of clouds in the A-band from a hand-held instrument by astronauts on the Gemini 5 mission in 1965 (Saiedy et al., 1967).





The following years saw significant milestones in theoretical understanding of the A-band and retrieval possibilities (e.g. McClatchey et al., 1973; Wu, 1985; Fischer and Grassl, 1991; O'Brien and Mitchell, 1992; Kuze and Chance, 1994) together with instrument development (e.g. Curran et al., 1981; Fischer et al., 1991; Asano et al., 1995). The first autonomous (i.e. non-astronaut operated) measurements of the A-band used for cloud altitude retrieval appear to be from the Kosmos 320 satellite in 1970 (Gorodetsky et al., 1971; Malkevich, 1973). The 1967 Kosmos 149 mission also had this measurement capability
(Malkevich, 1973; Marchuk et al., 1988), though those data appear not to have been used for cloud altitude retrieval, possibly due to on-orbit platform orientation issues with that satellite (Harvey and Zakutnaya, 2011). Both missions lasted several weeks. Either way, the launch of the Global Ozone Monitoring Experiment (GOME) in 1995 provided the first large-scale satellite measurements in the A-band used for generating cloud retrieval data sets (Koelemeijer et al., 2001; Loyola, 2004). GOME was itself a descoped version of the SCanning Imaging Absorption spectroMeter for Atmospheric CHartographY (SCIAMACHY;
Burrows et al., 1995), which was launched in 2002.

      The applicability of the A-band for cloud and aerosol altitude remote sensing is now well-established, as well as the weaker B-band near $687\,\mathrm{nm}$ (Desmons et al., 2019) and $O_2$-$O_2$ collision complex absorption near $477\,\mathrm{nm}$ (for high spectral resolution instruments; Acarreta et al., 2004). Both theory (e.g. Stephens and Heidinger, 2000; Heidinger and Stephens, 2000; Rozanov and Kokhanovsky, 2004; Yang et al., 2013b; Davis et al., 2022) and practical application (e.g. Vanbauce et al., 1998; Lindstrot
et al., 2006; Wang et al., 2008; Lelli et al., 2012; Richardson et al., 2019; Compernolle et al., 2021) demonstrate that A-band information content is a function of spectral characteristics (sampling and resolution), with key factors contributing to retrieval uncertainty being knowledge of the cloud vertical extinction profile (and particularly multi-layered cloud systems) and, for optically-thin clouds, surface reflectance characteristics. Many further references could be provided - the above are generally illustrative of the history and principles for different sensors sampling these features.

With some exceptions, instruments tend to fall into one of two categories. The first (e.g. GOME, SCIAMACHY) are spectrally fine ($<1\,\mathrm{nm}$) and spatially fairly coarse (several to hundreds of $\mathrm{km}$ pixel length), which provides more information on cloud vertical structure at the expense of errors in spatially-heterogeneous scenes. Those in the second category (including PACE's OCI) have coarser spectral sampling and resolution but spatially finer resolution, with the converse strengths and limitations. Additional discussion of the history of scientific understanding of $O_2$ absorption features and use for remote sensing
is provided by Davis et al. (2022).

      This paper presents an algorithm, Cloud Height Retrieval from $O_2$ Molecular Absorption (CHROMA), for determination of cloud altitude from future PACE OCI measurements. Radiative transfer (RT) simulations are used to understand sensitivities and expected performance. Simulated retrievals are also performed for the Ocean and Land Colour Instrument (OLCI), the first of which launched in 2016, and the potential utility of OLCI as an OCI proxy for CTP retrieval is demonstrated. The application
to real OLCI measurements will be shown in a follow-up study. Section 2 introduces the OCI and OLCI instruments, Section 3 describes the proposed retrieval algorithm, and Section 4 shows simulated retrievals for both instruments. Finally, Section 5 provides a summary of expectations for eventual PACE OCI data.





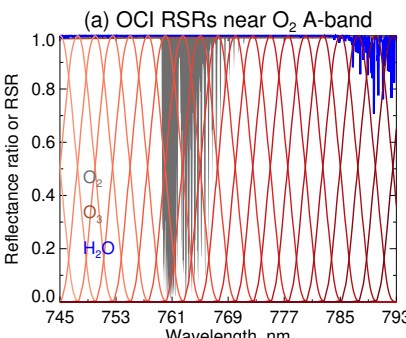 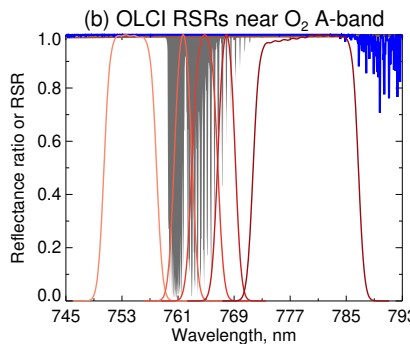

**Figure 1.** Relative spectral response functions (RSRs) for (a) OCI, based on current expectations and (b) Sentinel 3a OLCI, based on pre-launch measurements, in the region of the $O_2$ A-band. RSRs are shown in red, with different tones indicating different channels. Absorption by various species (see text) is shown in dark grey shading for $O_2$, brown lines for $O_3$ (which has very weak absorption), and blue lines for $H_2O$.

## 2   Sensor characteristics

OCI and OLCI share several similarities that suggest the latter would be a good proxy for pre-launch testing of a CTP retrieval
algorithm for the former. OCI (Werdell et al., 2019) will have continuous spectral coverage from approximately 350 to 890 nm, with channels spaced every 2.5 nm. They will be somewhat overlapped as the full width at half maximum (FWHM) will be 5 nm. In addition to this, OCI will have discrete channels (of various widths) centred near 940, 1040, 1250, 1378, 1620, 2130, and 2250 nm. These NIR and SWIR channels generally mirror those on heritage missions. OLCI (Donlon et al., 2012) is an enhanced successor to the MEdium Resolution Imaging Spectroradiometer (MERIS, Rast et al., 1999) and has 21 channels
across the visible and near-infrared, including 5 in the vicinity of the $O_2$ A-band (denoted channels OA12-OA16). MERIS lacked OLCI's channels centred near 764 and 767.5 nm. Two OLCI instruments are in orbit at present, on the Sentinel 3A and 3B satellites. In this study, where relative spectral response (RSR) functions are illustrated or used in calculations, pre-launch RSRs from Sentinel 3A OLCI are used (sensor mean RSR from all detectors).

RSRs for OCI and OLCI in this region are shown in Figure 1 in red. The other colours serve to highlight absorption features
of relevance. These figures were generated by running the libRadtran RT code version 2.0.4 (Emde et al., 2016) with its built-in REPTRAN trace gas absorption package (Gasteiger et al., 2014), with individual absorbers (here $O_2$, $O_3$, and $H_2O$) switched sequentially on and off. The figure illustrates gas absorption strengths by plotting the ratio of top of atmosphere (TOA) reflectance with a particular absorber switched on to that with it switched off. More specifically, this simulation was run using libRadtran's implementation of the Discrete Ordinates Radiative Transfer (DISORT) solver (Buras et al., 2011) with 32
hemispherical streams using a US standard atmosphere for a solar zenith angle of 45°, viewing zenith angle of 30°, and relative azimuth angle of 0°, over a Lambertian surface with spectrally-flat albedo of 0.2. The calculation was done at REPTRAN's fine-resolution setting of 1 cm$^{-1}$ ($\sim$0.05 nm in this spectral region). Both sensors sample window (i.e. negligible absorption) regions around the A-band and have several channels probing differing absorption strengths within the band. While OLCI



has fewer channels than OCI, those within the A-band are narrower, and well-placed to capture spectral regions of high and
moderate absorption. $O_3$ absorption is fairly spectrally flat and weak (two-way transmittance is approximately 0.99); $H_2O$
absorption becomes important from 785 nm onwards.

Besides the above spectral considerations, the sensors share similar spatial characteristics. OCI will be a single sensor with
a horizontal pixel size of approximately 1 km at the sub-satellite point and a swath width of 2,663 km. OLCI has 5 separate
cameras with a combined swath width of 1,270 km; native horizontal pixel size is approximately 300 m, although, as with
MERIS, a 'reduced resolution' aggregated data product at 1.2 km is also produced. This also means processing OLCI data
should yield sufficient statistics to assess the retrievals against other satellite products and ground truth prior to PACE launch.
Both sensors also tilt to decrease the proportion of the swath affected by strong Sun glint over water: OLCI is tilted 12.6°
westwards, OCI will tilt 20° north in the northern hemisphere and 20° south in the southern hemisphere, reversing near the
Equator.

## 3 Retrieval algorithm

### 3.1 Theoretical basis

As described in the Introduction, the factors influencing information content and error for cloud retrievals in the $O_2$ A-band are
well established. This section serves to provide some illustrative examples for OCI channels. Figure 2 shows TOA reflectance
integrated for the OCI channels in Figure 1a, for a variety of liquid cloud and surface conditions (same geometry and RT setup
as in Figure 1a, except as indicated in the caption). As COT increases (Figure 2a), TOA reflectance increases across the whole
spectral region. Outside of the A-band the spectrum remains fairly flat; channels sampling the A-band brighten but retain a dip
relative to the surrounding windows. By comparing Figures 1a and Figure 2, it can be seen that OCI's 5nm FWHM smooths
the two wings of the A-band into a single feature. Figure 2c shows that the spectral signature of an increasing surface albedo
is similar to that of increasing COT, meaning a constraint on surface albedo is necessary for a successful retrieval.
Figure 2b shows that as CTH increases (CTP decreases) the TOA signal in the window regions does not change. Channels
within the A-band become brighter for higher clouds, as more of the $O_2$ absorption is shielded. A smaller brightening is seen
for channels at wavelengths longer than 785 nm due to a similar shielding effect from $H_2O$ absorption. Finally, Figure 2d
shows that the spectral signature of changes in cloud vertical extent (here, the fraction of the column from cloud top to surface
occupied by the cloud) is very similar to that of CTH, with shallower clouds being brighter (as a higher effective scattering
altitude has a stronger effect of shielding the absorption from lower altitudes). This points to the need for an assumption or joint
retrieval of parameters relating to cloud vertical structure - in principle not just the cloud vertical extent, but also accounting
for variations in water content or CER (not shown here). Recently, Fischer and Preusker (2021) found that OLCI's A-band
measurement capabilities provide roughly two degrees of freedom for clouds in most cases - COT and CTP - slightly fewer
over deserts, and slightly more (partial information on vertical extent) in parts of the tropics. The information on vertical extent
comes from the enhancement of absorption by within-cloud multiple scattering; this enhancement varies across the A-band as
the cloud signal is spectrally flat but $O_2$ absorption is not (Kokhanovsky and Rozanov, 2004). Despite OCI's greater number





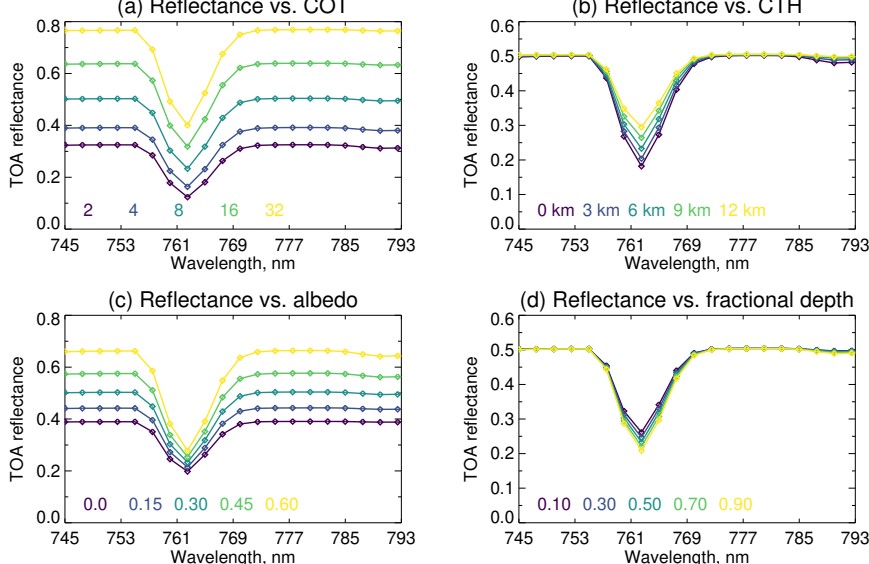

**Figure 2.** TOA reflectance for selected OCI channels (diamonds) as a function of cloud and surface properties. The base simulation is for a liquid water cloud with CER 10 μm and midvisible COT of 8. The CTH is 6 km and the fractional depth (i.e. fraction of the altitude between the CTH and ground level) of the vertically-homogeneous cloud is 0.5 (i.e. cloud base of 3 km, cloud geometric thickness of 3 km). Simulations also include Rayleigh scattering and a spectrally-flat Lambertian surface with albedo 0.3. All simulations otherwise as described for the setup used for Figure 1. Panels show the variation in TOA reflectance when (a) COT, (b) CTH, (c) surface albedo, or (d) fractional depth are varied and all other parameters held constant. Coloured parameter values at the bottom of each panel correspond to the coloured lines.

of spectral channels, their comparative broadness and overlap means that parameters related to cloud vertical structure are also not likely robustly retrievable from OCI's A-band measurements and instead must be assumed.

The proposed CHROMA algorithm strategy, then, is to simultaneously retrieve COT, CTP/CTH, and surface albedo (with
a strong prior constraint on the latter) from OCI or OLCI measurements within the A-band. Other atmospheric and surface properties will be assumed or obtained from meteorological reanalyses. The choice of whether to retrieve or fix surface albedo is an important one. If allowed to vary totally freely the retrieval is more likely to converge to the incorrect solution as the spectral signatures of clouds and the surface are similar so there can be ambiguity. On the other hand, fixing albedo can lead to biases in altitude if the fixed value is wrong - particularly for optically thinner clouds. Section 5.6 of Preusker and Fischer
(2021) illustrates this for clouds, and Section 2 of Sanders et al. (2015) analyses this issue from the point of view of aerosol layer height retrieval, which is analagous to the case of low-COT clouds. The goal is therefore to obtain a strong prior constraint but allow for some flexibility to account for the fact the prior is not perfect.

For OCI, the eight consecutive channels centred from 755 to 772.5 nm will be used. For OLCI, the four channels OA12-OA15 (nominal centres 753.75, 761.25, 764.375, and 767.5 nm) will be used. The rationale for excluding other window chan-





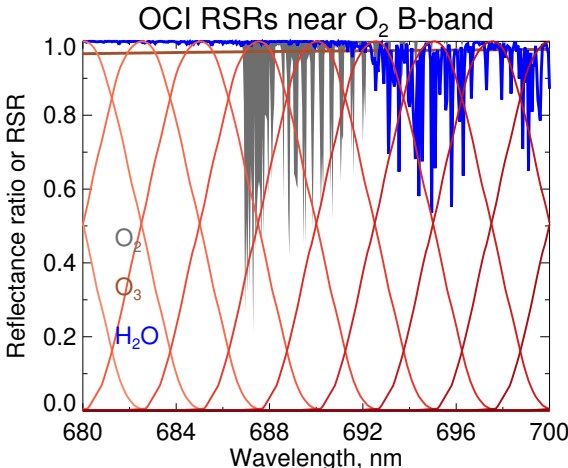

**Figure 3.** As Figure 1, except for OCI channels in the region of the O$_2$ B-band.

nels (several options for OCI, and OA16 for OLCI) is that these will be essentially redundant, given almost identical response to changes in geophysical parameters, and (shown later) very strong spectral correlations in forward model uncertainty. At cloud radiance levels, sensor noise is negligible and sensor absolute calibration uncertainty is likely to be spectrally flat (and so highly correlated, e.g. Neneman et al., 2020). There is, however, a benefit in retaining an overall shorter spectral range in the retrieval as it decreases uncertainties related to spectral variation in surface and aerosol properties across the channels used.

The reasons for simultaneous retrieval of COT, CTH, and albedo as opposed to taking COT from the separate MODIS-like optical properties retrieval and retrieving only CTH and albedo are two-fold. The first reason is that the optical properties retrieval requires CTH as input in order to calculate trace gas absorption corrections for the channels it uses (Platnick et al., 2021), so the CTH retrieval needs to come first. The second is that using input from a different COT retrieval with different assumptions makes uncertainty quantification and propagation difficult. An alternative approach would be to perform a simul-

taneous retrieval of all cloud parameters - such retrievals exist (e.g. Poulsen et al., 2012), but an advantage of the approach in this study is that it allows greater continuity with the MODIS/VIIRS processing chain, is computationally simpler, and reduces risk of not having an algorithm in place by PACE launch.

As discussed earlier, the O$_2$ B-band has also been used for retrieval of cloud and aerosol altitude and in principle a retrieval could benefit from including additional absorption features. Figure 3 shows OCI RSRs in this region; relative to Figure 1, the

B-band is weaker and narrower, and OCI channels in this region will be more strongly influenced by O$_3$ and H$_2$O absorption. O$_2$-O$_2$ dimer absorption (not shown) near 477 nm is weaker still, while O$_3$ and aerosol signals are stronger, and NO$_2$ absorption can also be significant. Thus, use of only the A-band spectral region entails fewer necessary assumptions or constraints on surface and aerosol properties (and H$_2$O profiles). While H$_2$O absorption could in principle be used for CTP retrieval in a similar way to O$_2$, the fact that H$_2$O column amount and profile shape vary strongly while O$_2$ is well-mixed is a disadvantage.



For these reasons only the A-band channels are considered in the present approach. OLCI also lacks observations in these other spectral regions, limiting the ability to test an algorithm combining A- and B- bands before PACE's launch. While the Earth Polychromatic Imaging Camera (EPIC) sensor has measurements in both spectral regions (see Yang et al., 2013b; Davis et al., 2022, for cloud retrievals using both bands), its significantly coarser pixel size (approximately 8 km across at nadir) means that scene heterogeneity is a bigger issue than for OCI. Additionally, EPIC's orbit (at the L1 Lagrange point) vs. OCI

and OLCI's Sun-synchronous polar orbits mean that solar and viewing geometry characteristics are very different. As a result OLCI seems the better proxy for OCI. Combining $O_2$ A- and B-band channels would however, be useful in a retrieval of aerosol properties as spectral surface reflectance would need to be characterised anyway, and land surface reflectance is often darker around the B-band than A-band. Xu et al. (2019) demonstrated this type of retrieval for EPIC.

## 3.2    Radiative transfer

### 3.2.1    Spectral setup


All RT calculations in this study are performed with the DISORT solver of libRadtran version 2.0.4 (Emde et al., 2016), with 32 streams per hemisphere and a pseudospherical atmosphere (direct solar beam is treated in spherical geometry, scattered light is plane-parallel). The fundamental quantity calculated by this is the monochromatic TOA reflectance $\rho(\lambda)$ at wavelength $\lambda$,

$$\rho(\lambda) = \frac{\pi D_\odot^2 L(\lambda)}{\mu_0 F_0(\lambda)},  \tag{1}$$

where $D_\odot$ is the ratio of the Earth-Sun distance to 1 astronomical unit (AU), $L$ the TOA radiance, $F_0$ the downwelling solar spectral irradiance at TOA, perpendicular to the Sun and at 1 AU, and $\mu_0$ the cosine of the solar zenith angle. In this study, $\rho(\lambda)$ is calculated with a sampling of 0.01 nm over the wavelength range 748-782 nm, encompassing all relevant portions of the OCI and OLCI RSRs. Then, the TOA reflectance of OCI or OLCI channel $i$, $\rho_i$ is obtained:

$$\rho_i = \frac{\int_{748}^{782} \rho(\lambda) F_0(\lambda) \Phi_i(\lambda) \, d\lambda}{\int_{748}^{782} F_0(\lambda) \Phi_i(\lambda) \, d\lambda}  \tag{2}$$

Here, $\Phi_i(\lambda)$ indicates the RSR of sensor channel $i$. The factor $F_0(\lambda)$ appears in both Equations 1 and 2 because the RSR is defined for spectrally uniform (white) light while the sensor measures total radiance across the channel, so weighting is necessary. The $F_0$ spectrum used is from Coddington et al. (2021); this is at 0.025 nm spacing with 0.1 nm bandwidth, and is linearly interpolated to the 0.01 nm grid used for RT calculations in this study. It has 0.3 % uncertainty in this spectral range. Numerically, trapezoid integration is used for Equation 2; the spectrally-integrated $\rho_i$ are used as input to the retrieval

algorithm.

### 3.2.2    Atmospheric profile and gas absorption

The atmospheric temperature/pressure/height ($T/p/z$) profile used is a 20-level (19-layer) 1976 US Standard Atmosphere (Dubin et al., 1976), with layer bounds 0, 1, 2, 3, 4, 5, 6, 7, 8, 10, 12, 14, 17, 20, 25, 30, 40, 50, 70, and 100 km. libRadtran takes input in height ($z$) space; conversions between $z$ and $p$ are done in $\log_{10}(p)$ space (as this varies close to logarithmically with





height). Pressures at the lowest and highest bounds are 1013 and 0.00032 mb, respectively. Where calculations are performed for a different surface pressure, the whole $p$ profile is simply scaled accordingly; this creates a minor error source, as Rayleigh scattering and gas absorption are slightly dependent on $T$ as well as $p$, and in reality for a lower surface pressure (e.g. elevated terrain) one might expect $T$ to also be cooler.

Note that the key quantity relevant for the radiative transfer here is pressure: essentially, there is a single (global) $p$-$T$
relationship assumed at the radiative transfer stage. While this relationship varies in the real world, often as a function of latitude, it is not computationally practical to use multiple basic profiles in retrieval processing, and in any case $T$ is a secondary error source. Since the instrument and retrieval sensitivity is to $p$ rather than $z$, in retrieval processing the solution will be obtained in $p$ space and then transformed to $T$ and $z$ (for output CTT and CTH) with the aid of ancillary reanalysis or forecast profiles. In this work the US standard atmosphere is also used for this conversion in the simulated retrievals.

Trace gas absorption calculations use the recently-released HITRAN 2020 data base (Gordon et al., 2022). $O_2$ absorption is calculated assuming a volume mixing ratio of 0.21, performing line by line calculations with a Voigt line shape and cutting off $25 \, \mathrm{cm}^{-1}$ from line centres (as customary, e.g. Clough et al., 1989). While REPTRAN (Gasteiger et al., 2014) was used in earlier figures for illustrative purposes, for CHROMA the newer data from HITRAN 2020 (and running RT calculations at 0.01 nm spacing) are preferred, for increased accuracy at the cost of (considerably) more calculation time. $O_3$ is added using the 295 K
spectrum of Bogumil et al. (2003) assuming 300 DU in the uppermost atmospheric layer. While this treatment is fairly simple, $O_3$ absorption is weak in this spectral region (nadir optical thickness $\sim$0.0033) that the error introduced by variations in $O_3$ amount and profile structure are negligible. Other absorbers are neglected as they are negligible in this spectral region. Finally, $O_2$ self- and foreign continua from version 3.4 of the MT_CKD model (Mlawer et al., 2012) are added. Note this version of MT_CKD was developed using an older version of the HITRAN line data base (as continuum models are semi-empirical,
based on fitting residual absorption after removing known gas lines), creating a minor inconsistency. Prior to application to real OCI data, updates will be made to absorption calculations as necessary.

Rayleigh optical depth is calculated using Bodhaine et al. (1999), and is $\sim$0.027 in the A-band for a surface pressure of 1013 mb.

### 3.2.3 General cloud setup

libRadtran includes precalculated liquid and ice cloud phase matrices and scattering/extinction properties, used here, tabulated based on lognormal distributions with various effective radii. Clouds are assumed to be single-layered and single-phase (i.e. consisting solely of either liquid water droplets or ice crystals).

The cloud layer is divided into 5 contiguous sublayers, running from the CTH to the cloud base height (CBH). Cloud fractional geometric depth (FGD) is then defined as 1-(CBH/CTH). The use of sublayers allows for vertical variation of
cloud structure. If a cloud boundary does not align exactly with an atmospheric layer, libRadtran preserves the $T/p/z$ profile and calculated atmospheric scattering/absorption and inserts new layers to allow clouds (or aerosols) to be precisely at user-specified altitudes. Thus, in most cases, the final calculation is done with more than the 19 layers discussed in Section 3.2.2.





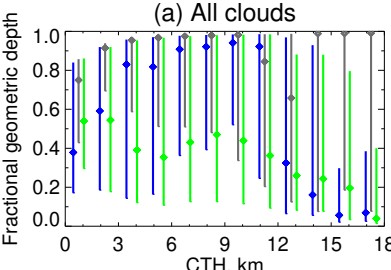
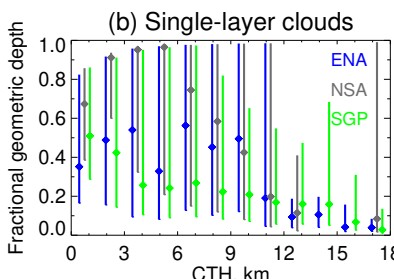

**Figure 4.** Median (points) and central one standard deviation (lines) of cloud FGD binned as a function of CTH for ARSCL data from three ARM sites. Panels show results for (a) all clouds and (b) single-layer clouds. Only bins with at least 1000 observations are populated. Colours indicate the ENA (blue), NSA (grey), and SGP (green) sites. The bin size is 1.5 km; sites are slightly offset from bin centres for clarity.

As mentioned earlier, parameters relating to cloud vertical structure will not be directly retrieved so some assumptions are necessary. To guide these assumptions (discussed in Sections 3.2.4 and 3.2.5), cloud FGD as a function of CTH is shown
for three Atmospheric Radiation Measurement (ARM) sites in quite different environments in Figure 4. These sites are East North Atlantic (ENA; 39.1° N, 28.0° W), North Slope of Alaska (NSA; 71.3° N, 156.6° W), and Southern Great Plains (SGP; 36.6° N, 97.5° W). These plots are constructed from four years (2017-2020) of the Active Remote Sensing of Cloud Locations (ARSCL) product (Kollias et al., 2016), and shown separately for all clouds and single-layer clouds. Unfortunately, FGD shows high variability between locations and within CTH bins at a single location, but cloud vertical extent is difficult
to retrieve from A-band observations at these spectral resolutions (Fischer and Preusker, 2021). Note that most cloud property retrieval algorithms from passive sensors assume vertically-homogeneous clouds, sometimes with minimal vertical extent.

### 3.2.4 Liquid phase cloud properties

CHROMA assumes a cloud top CER of 11 μm for liquid clouds, which is close to the global median and modal values from MODIS retrievals (King et al., 2013). The CBH is assumed to be half the CTH (i.e. FGD=0.5) based on Figure 4. Within the
cloud, vertical structure of CER and liquid water content (LWC) follow the adiabatic model (e.g. Brenguier et al., 2000). That is, LWC in each sublayer decreases linearly as one descends from CTH to CBH. The CER of each sublayer also decreases descending from the cloud top, proportional to $(h/h_0)^{1/3}$ where $h/h_0$ represents the geometric fraction of the cloud remaining below the sublayer height (i.e. $h/h_0$ is 1 at cloud top and 0.2 for the lowest of the 5 sublayers). This gives a cloud top CER of 11 μm and a cloud base CER of $11 \times 0.2^{1/3} \approx 6.4$ μm. Real clouds are often subadiabatic; however, in these cases, vertical
variation of LWC often remains close to linear (albeit at a different rate) and variation of CER is similar to that for an adiabatic cloud (Boers et al., 2006; Brenguier et al., 2000). Note that libRadtran is provided only with the shape of the LWC profile, and the per-layer mass is scaled as required to obtain the total COT of the cloud. As such this vertical structure of LWC and CER is likely reasonable for subadiabatic clouds as well.





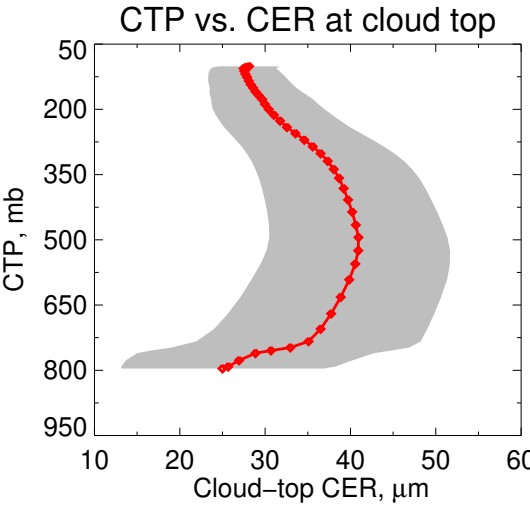

**Figure 5.** Ice cloud top effective radius (CER) vs. cloud top pressure (CTP) relationship used in this study. The red line denotes the mean profile, and grey shading 1 standard deviation around the data collected by van Diedenhoven et al. (2020).

### 3.2.5 Ice phase cloud properties

For ice phase clouds, the severely roughened solid column aggregates model from Yang et al. (2013a) is used. This habit is also used in the latest MODIS cloud optical properties retrieval (Platnick et al., 2017). The severely roughened model is used (smooth and moderately-roughened are also available) as, except for the warmest ice clouds, high levels of shape distortions are nearly ubiquitous (Yang et al., 2013a; van Diedenhoven et al., 2020, and references therein). The FGD is assumed to be 0.25 (Figure 4).

The cloud-top CER is assumed to be a function of CTP, using a data set collected by van Diedenhoven et al. (2020). For ice crystals the effective radius is defined as $\frac{3V}{4A}$ where V is the bulk volume (total mass divided by bulk ice density) and A is the mean projected area assuming random orientation of crysyals. van Diedenhoven et al. (2020) collated a year of MODIS retrievals for ice clouds with COT>1, and binned them as a function of CTT as theory and observations of convective ice clouds link this (among other factors) to crystal growth rate and thus size. The average CTP within each CTT bin was also

calculated. These data were transformed to give the CTP-CER relationship in Figure 5. The standard deviation within each bin varies from about $3.5\,\mu m$ for the coldest ice clouds to $12\,\mu m$ for the warmest, and decreases somewhat if only clouds with COT>3 are considered (not shown), potentially due to increased retrieval uncertainty for optically-thinner clouds. A large proportion of the variability is due to latitudinal variation (van Diedenhoven et al., 2020) - as noted earlier, it is unfortunately only computationally feasible at present to incorporate a single relationship for retrieval processing.

Within the cloud, vertical structure of CER is based on the results for convective ice clouds by van Diedenhoven et al. (2016). This is determined by each sublayer's altitude with respect to the homogeneous freezing level (HFL, where the temperature first reaches 239 K), which is around $7.5\,km$ (and $380\,mb$) for the US standard atmospheric profile. For sublayers beginning from





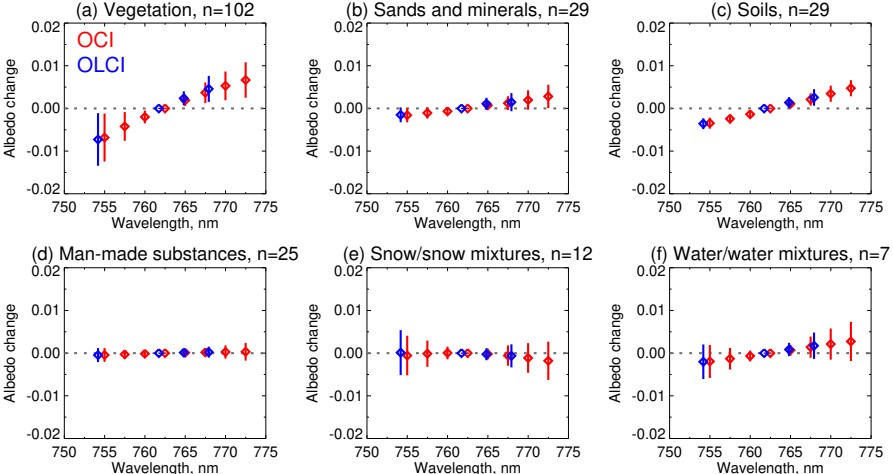

**Figure 6.** Spectral slope of surface albedo across the A-band, for (red) OCI and (blue) OLCI, expressed as differences relative to the albedo at the sensor channel of maximum absorption. Points and lines show the mean and standard deviation of spectral differences in each category in each category. Panels show (a) vegetation, (b) sands and minerals, (c) soils, (d) man-made substances, (e) snow and snow mixtures, and (f) water and water mixtures. The number of spectra in each category is given in each panel's title.

cloud top down to the HFL, the CER increases as height decreases by $3\,\mu\mathrm{m}\,\mathrm{km}^{-1}$. For those below the HFL, CER increases as height decreases by $6\,\mu\mathrm{m}\,\mathrm{km}^{-1}$. Vertical profiles of ice water content (IWC) are assumed to be linearly increasing as height

descends from CTH to CBH; note this is essentially the opposite of the adiabatic LWC profile. While real IWC profiles can vary enormously, this general pattern is commonly found in both airborne (Krämer et al., 2020) and satellite observations (Feofilov et al., 2015), especially for optically-thicker clouds.

### 3.2.6 Aerosol properties

Aerosols are included in the simulation with the Shettle (1989) 'rural' type up to $2\,\mathrm{km}$, the 'background' type above that, and

an aerosol optical thickness (AOT) of 0.14 at $550\,\mathrm{nm}$ (approximately 0.1 in the A-band). This is within the range of typical background values retrieved by many satellite data sets (e.g. Sogacheva et al., 2020) Note that aerosol loading is varied in the sensitivity tests later, but in general (except for cirrus clouds) cloud scattering is an order of magnitude or more stronger than aerosol so (with some regional exceptions) sensitivity to aerosol assumptions is low.

### 3.2.7 Surface properties

For computational simplicity the surface (retrieved with a strong prior constraint) is assumed to be Lambertian, i.e. described by an isotropic albedo rather than a bidirectional reflectance distribution function (BRDF). This is justified as, particularly for optically-thicker clouds, much of the light will be multiply scattered and so the light field will have lost some directionality. In addition, the surface is taken to be spectrally flat across this wavelength range. This assumption is evaluated in Figure 6,



using data from the ECOsystem Spaceborne Thermal Radiometer Experiment on Space Station (ECOSTRESS) spectral library
(Baldridge et al., 2009; Meerdink et al., 2019). This contains reflectance spectra of several thousand substances (natural and
man-made), made by several instruments which together cover the visible and TIR spectral ranges. Importantly, to minimise
change of appearance, vegetation samples (cuttings) are measured rapidly after acquisition. Some measurements in the data
base are bidirectional reflectance (at a single angle) while others are hemispherical directional reflectance. This is expected to
have negligible influence for the present purpose (i.e. spectral shape over a narrow region).

A total of 204 spectra containing data in the A-band region were manually classified and convolved with OCI and OLCI
spectral response functions for the channels used in CHROMA. These are taken as representative of the surface albedo that
would be seen by these sensors - although at satellite pixel scales, the real surface would likely consist of mixtures of multiple
types. Then, difference spectra were calculated by subtracting the albedo at the most-absorbing satellite channel (centred near
762.5 nm for OCI and 761.7 nm for OLCI). The mean and standard deviation of these difference spectra within each category
are shown in Figure 6. Half of the spectra correspond to vegetation of various types; these were additionally divided into
several categories (grasses, trees, shrubs, and mixed samples) but are combined in the figure, as it was found that the slopes for
different categories of vegetation are similar.

    From Figure 6, the typical vegetation albedo increases by about 0.02 across the A-band; for sands and soils the increase
is around 0.01. It is therefore likely that most vegetation-covered satellite pixels would show a similar increase. Man-made
substances (which included various types of asphalt, concrete, brick, wood, metal, and plastic) are very spectrally flat. The
'snow' category included snow samples of various ages and snow-vegetation mixtures; these difference spectra are also, on
average, flat. The 'water' category (which includes coastal, sea, wetland, and other samples) can show a small positive increase
across the band. Note that the open ocean (about 2/3 of the Earth's surface) is both dark and quite spectrally flat due to strong
absorption by water in this spectral region (e.g. Hale and Querry, 1973), decreasing TOA sensitivity to assumptions. As a result
of the above, the assumption of a spectrally flat albedo for computational simplicity seems justified (although the effects of the
assumption are tested later).

### 3.3 Optimal Estimation solution

### 3.3.1 Retrieval formalism

Optimal Estimation (OE) is a widely-used Bayesian technique that aims to find the maximum *a posteriori* (MAP) solution to
the retrieval given measurements, *a priori* information on the state variables, and any additional constraints available (Rodgers,
2000). One advantage of the technique is that it, with some assumptions and caveats (e.g. Povey and Grainger, 2015), returns
an estimate of the uncertainty on the retrieved state for each individual retrieval.

    OE algorithms like CHROMA require three input vectors, a forward model to relate them, and two main matrices. The
measurement vector $\boldsymbol{y} = (\rho_0, \rho_1, \ldots, \rho_n)$ consists of observed TOA reflectances in $n$ channels; eight for OCI or four for OLCI.
The state vector $\boldsymbol{x}$ consists of the retrieved quantities (here COT, CTP, and surface albedo) and is matched by a vector of *a
priori* values for those state parameters, denoted $\boldsymbol{x}_a$. For satellite data processing, surface albedo priors could be taken from the





**Table 1.** Radiative transfer LUT node points, for each satellite channel, in the CHROMA algorithm.

| Dimension | Node points |
|---|---|
| Solar zenith angle (°) | 0, 5, 10, …, 80 |
| Viewing zenith angle (°) | 0, 4, 8, …, 72 |
| Relative azimuth angle (°) | 0, 9, 18, …, 180 |
| $\log_{10}$(COT) | -0.5, -0.167, 0.167, …, 2.167 |
| CTH (km), liquid phase | 0.3, 1, 2, 3, 4, 5, 6, 7 |
| CTH (km), ice phase | 3, 5, 7, 9, 11, 13, 15, 17 |
| Surface albedo | 0, 0.1, 0.2, …, 1 |
| Surface pressure (mb) | 1050, 900, 750, 600 |

MERIS database described by Popp et al. (2011). Section 4 discusses prior albedo values for the simulated retrievals presented later in this study. No prior constraint is applied to COT or CTH in the algorithm. The forward model is an approximation of the true forward function (the real propagation of radiation in the cloudy Earth-atmosphere system). For computational

tractability this is a lookup table (LUT) of RT code results. The forward model $\mathbf{F}$ operates on the state vector to return simulated measurements ($\boldsymbol{y}_{LUT}$) from the LUT, i.e. $\boldsymbol{y}_{LUT} = \mathbf{F}(\boldsymbol{x})$, using multiple linear interpolation. There is one LUT file for each cloud phase (ice or water) with dimensions for each satellite channel given in Table 1. Note the $\log_{10}$(COT) bounds translates to an absolute COT range of 0.5 to about 150.

The two necessary matrices describe uncertainty covariances, under the assumption that uncertainties are unbiased and

Gaussian. The matrix $\mathbf{S}_y$ describes the measurement and forward model uncertainties; the latter is sometimes split into a separate matrix denoted $\mathbf{S}_b$, although the result is numerically and mathematically equivalent if (as here) both contributions are contained within $\mathbf{S}_y$. Estimation of $\mathbf{S}_y$ is, in general, complicated and is the topic of Section 3.3.2. The matrix $\mathbf{S}_a$ describes the uncertainty of the *a priori* values $\boldsymbol{x}_a$. This is assumed to be diagonal, and COT and CTH are unconstrained within the bounds of LUT values (i.e. diagonal elements of the covariance matrix are an arbitrarily large number). Surface albedo is more tightly

constrained, with a one standard deviation uncertainty of $\pm 0.01$ over water and $\pm 0.05$ over land or snow. These constraints can be re-evaluated following the application to real data.

The MAP solution is obtained by minimising a cost function,

$$J(\boldsymbol{x}) = (\boldsymbol{y} - \boldsymbol{y}_{LUT}(\boldsymbol{x}))^{\mathrm{T}} \mathbf{S}_y^{-1} (\boldsymbol{y} - \boldsymbol{y}_{\mathbf{LUT}}(\boldsymbol{x})) + (\boldsymbol{x} - \boldsymbol{x}_{\mathbf{a}})^{\mathrm{T}} \mathbf{S}_a^{-1} (\boldsymbol{x} - \boldsymbol{x}_{\mathbf{a}}), \tag{3}$$

where the first term represents squared deviations between the observations and forward model, and the second between the

state vector and *a priori* values, in both cases weighted by their covariances. The first guess at the solution the LUT node point (for the appropriate geometry and surface pressure; Table 1) with the lowest cost $J$. Levenberg-Marquardt iteration (Levenberg, 1944; Marquardt, 1963) is used to find the minimum of Equation 3 to determine the retrieved state $\hat{\boldsymbol{x}}$; the retrieved is deemed to have converged when the decrease in $J$ between successive iterations is smaller than 0.01 (typically 5 or fewer iterations). Over a large ensemble of retrievals, distributions of $J$ should approach a $\chi^2$ distribution, although for cases where off-diagonal





elements of $\mathbf{S}_y$ and/or $\mathbf{S}_a$ are significant and may change from one retrieval to another (such as here) the degrees of freedom of the distribution may be smaller than expected based on the number of measurements and state variables. Unusually high values of $J$ indicate retrievals with a poor level of fit between the measurements and forward model at the retrieval solution. Extremely high $J$ can be used as a measure of quality control as it indicates that it is likely that some aspect of the forward model or uncertainty has been misspecified for the retrieval in question.

Assuming that the forward model is appropriate and unbiased, that covariance matrices $\mathbf{S}_y$ and $\mathbf{S}_a$ are of appropriate magnitudes, that the retrieval converges to the global minimum of the cost function, and that the forward model is close to linear near the solution (Povey and Grainger, 2015), then the covariance matrix $\hat{\mathbf{S}}_x$ describing the uncertainty on the retrieved state can be calculated (e.g. Section 4.1 of Rodgers, 2000):

$$\hat{\mathbf{S}}_x = \left(\mathbf{K}^{\mathrm{T}}\mathbf{S}_y^{-1}\mathbf{K} + \mathbf{S}_a^{-1}\right)^{-1} \tag{4}$$

In the above $\mathbf{K}$ is the gradient of the forward model with respect to the state vector, i.e. $\partial\mathbf{F}/\partial\boldsymbol{x}$, also known as the Jacobian matrix or weighting function. Here, it is calculated numerically using the finite difference method from nearby LUT node points. The square root of the diagonal elements of $\hat{\mathbf{S}}_x$ provide the one standard deviation Gaussian uncertainties on the retrieved state, and the off-diagonal elements describe the covariance between the retrieved quantities. Since what is actually retrieved in the code corresponds to LUT indices, retrieved values and uncertainties on retrieved cloud altitude are easily

calculable in each coordinate system (i.e. CTH, CTP, CTT).

Note that as radiative transfer is nonlinear in most parameters, the assumption of linearity near the solution can break down; in these cases, the magnitude of $\hat{\mathbf{S}}_x$ can be incorrect. Vukicevic et al. (2010), Witek et al. (2018) and Western et al. (2020) present methods which avoid this assumption, essentially by testing to see to what extent each node in a retrieval LUT represents a plausible solution. This relies on a denser (more nodes) LUT than is practical for this specific retrieval problem.

Nonetheless, the uncertainty estimates provided by OE and other techniques can (and should) be evaluated to determine when they can be trusted; Sayer et al. (2020) and other references discussed therein present a method to do this, applied later to the simulated retrievals in Section 4.

If the cloud phase is unknown, then the retrieval is performed sequentially for both liquid and ice cloud models and the phase resulting in the lowest value of $J$ is chosen. For OCI, it is expected that a separate cloud phase algorithm will be available at

launch (Coddington et al., 2017) and used to determine the phase used. Propagation of discrete model choices through the OE formalism is less straightforward, and phase selection approaches (and related uncertainties) will be examined further following PACE's launch.

A final useful matrix is the averaging kernel $\mathbf{A}$, which describes the sensitivity of the retrieved state $\hat{\boldsymbol{x}}$ to changes in the true state $\boldsymbol{x}$ (e.g. Section 4.1 of Rodgers, 2000):

$$\mathbf{A} = \frac{\partial\hat{\boldsymbol{x}}}{\partial\boldsymbol{x}} = \hat{\mathbf{S}}_x\mathbf{K}^{\mathrm{T}}\mathbf{S}_y^{-1}\mathbf{K} \tag{5}$$

This essentially quantifies how strongly each of the measurement/forward model and prior uncertainties contribute to the covariance $\hat{\mathbf{S}}_x$ of the retrieved state. For the diagonal elements of $\mathbf{A}$, a value of 1 means that the retrieved state is constrained





entirely by the measurements $\boldsymbol{y}$, and a value of 0 that the measurements have provided no constraint and that state vector element was determined entirely by the *a priori* value. For CHROMA, COT and CTH are unconstrained by priors so the
relevant elements of $\mathbf{A}$ are always 1. For surface albedo, as COT increases the element of $\mathbf{A}$ corresponding to surface albedo approaches 0 because for an optically-thick cloud the contribution of the surface to the TOA signal is negligible. The trace of $\mathbf{A}$ gives the degrees of freedom for signal of the retrieval. Note that $\mathbf{A}$ is not a measure of the error in the retrieval, only the extent to which state vector elements are constrained by the measurements vs. prior.

### 3.3.2 Estimating the covariance matrix $\mathbf{S}_y$

$\mathbf{S}_y$ contains contributions from measurement uncertainty (noise and systematic) as well as forward model uncertainty. Its elements $i, j$ can be decomposed as

$$\mathbf{S}_{y,ij} = \sigma_{N,i}\sigma_{N,j}r_{N,ij} + \sigma_{S,i}\sigma_{S,j}r_{S,ij} + \sigma_{F,i}\sigma_{F,j}r_{F,ij} \tag{6}$$

where $\sigma$ indicates a Gaussian standard deviation uncertainty term, $r$ a channel-to-channel correlation, and subscripts $N$, $S$, and $F$ terms relating to radiometric noise, systematic calibration uncertainty, and forward model uncertainty respectively. The
former two terms are comparatively simple. Sensor noise is assumed to be 0.5 % of the TOA reflectance in each channel. This is higher than will be the case for real OCI data, but results are not sensitive to the precise number assumed as it is far outweighed by the other contributions. The noise term is also assumed spectrally uncorrelated, so $r_N$ is the Kronecker delta ($\delta_{ij} = 1$ if $i = j$, $0 \, \forall \, i \neq j$). The systematic calibration uncertainty is taken as 2 % of TOA reflectance, in line with OCI expectations (Werdell et al., 2019), and is assumed to be fully correlated across the spectral region ($r_S$=1), as is thought to be approximately
the case for OLCI (Neneman et al., 2020). This is a reasonable assumption because the spectral range used in the algorithm is small. Potentially, post-launch vicarious calibration (using e.g. the technique of van Diedenhoven et al., 2010) and on-board monitoring could decrease this uncertainty (and spectral correlation).

The forward model uncertainty component is both larger and more difficult to estimate accurately. It involves representing the uncertainty on the LUT of TOA reflectance used in the retrieval due to factors including physical approximations (i.e.
atmospheric and surface properties), LUT interpolation, and RT assumptions. Here, the approach is as follows:

1. Create an ensemble of simulated measurements $\boldsymbol{y}_\delta = (\rho_{\delta,0}, \rho_{\delta,1}, \ldots, \rho_{\delta,n})$ across the range of interest of state vector elements, but perturbing forward model assumed parameters (e.g. cloud vertical structure, surface spectral slope) within reasonable ranges.

2. Compare these to the LUT values for the equivalent state vector (i.e. interpolate the LUT for the values of COT, CTH,
and albedo for the given surface pressure, phase, and geometry; Table 1) to calculate the forward model error $\boldsymbol{y}_{LUT} - \boldsymbol{y}_\delta$ for each simulation.

3. Develop a simple parametric form to represent this uncertainty's magnitude $\sigma_F$ and its spectral correlation $r_F$ within Equation 6.





**Table 2.** Sampling of geophysical parameters for liquid cloud perturbation simulations. Fractions indicate the chance of an outcome for draws from discrete distributions.

| Quantity | Sampling distribution |
| --- | --- |
| | Geometry |
| Solar zenith angle (°) | $U(0, 80)$ |
| Viewing zenith angle (°) | $U(0, 72)$ |
| Relative azimuth angle (°) | $U(0, 180)$ |
| | Atmosphere |
| $\log_{10}(\text{AOT})$ at 550 nm | $N(-0.85, 0.2)$ |
| Surface pressure (mb) | 3/15 for 1013, 2/15 for each of 1000, 975, 950, |
| | 1/15 for each of 1025, 925, 900, 950, 750, 600 |
| | Surface |
| Surface reflection | 1/3 ocean-like Lambertian $U(0.0, 0.08)$, |
| | 1/3 land-like Lambertian $U(0.15, 0.65)$, |
| | 1/3 snow-like Lambertian $U(0.75, 0.95)$ |
| Surface spectral variation | Linear from 748 to 782 nm with difference $N(0, 0.015)$ |
| | Cloud |
| $\log_{10}(\text{COT})$ | $N(0.8, 0.5)$ truncated at -0.5 and 2 |
| $\log_{10}(\text{CER})$ at cloud top (µm) | $N(1.04, 0.15)$ truncated at CER of 5 and 25 |
| CER/LWC basic profile | 2/3 as in Section 3.2.4, 1/3 vertically homogeneous |
| CER gradient perturbation (µm) | $N(0, 3)$ additional linear change from top to base |
| Water content gradient perturbation | $N(0, 15\%$ of peak) additional linear change from top to base |
| CTH (km) | $U(0.5, 6)$ |
| FGD | $U(0.1, 0.9)$ |

Here, an ensemble of 90,000 perturbation simulations is used, with a 50 % probability of liquid or ice phase clouds for each. Sampling of geophysical parameters is shown in Tables 2 and 3 for the liquid and ice-phase simulations, respectively. $U(a, b)$ indicates random draws from a uniform distribution with minimum $a$ and maximum $b$, and $N(a, b)$ draws from a Normal (Gaussian) distribution with mean $a$ and standard deviation $b$. The choice of sampling distribution parameters is motivated by plausible variations of the quantities in question for single-layer cloud systems in various atmospheres and surface types.

Variation of COT, CTH, and cloud-top CER was informed by distributions of MODIS retrievals in King et al. (2013). Ice cloud habits were informed by Yang et al. (2013a) and Baum et al. (2014), and profiles and roughness by values and variation within van Diedenhoven et al. (2016, 2020) together with Krämer et al. (2020) and Figure 5. The ice crystal habits require one modification to step 2 above when interpolating the LUT to estimate forward model uncertainty (i.e. $\boldsymbol{y}_{LUT} - \boldsymbol{y}_{\delta}$). As the crystals are significantly larger than the wavelength of light, and their absorption is negligible in the visible, their





**Table 3.** Sampling of geophysical parameters for ice cloud perturbation simulations. Quantities not given here are as in Table 2.

| Quantity | Sampling distribution |
| --- | --- |
| Ice crystal habit | 6/10 8-element column aggregates, 1/10 for each of hollow bullet rosettes, solid bullet rosettes, droxtals, and 5-element horizontal plate aggregates |
| Ice crystal roughness | 2/3 severe, 1/3 moderate |
| CER at cloud top (μm) | Figure 5 with perturbation $N(0,7)$ |
| CER/IWC basic profile | 2/3 as in Section 3.2.5, 1/3 vertically homogeneous |
| CER gradient perturbation (μm) | $N(0,4)$ additional linear change from top to base |
| CTH (km) | $U(4,16)$ |
| HFL (km) | $N(7.5, 1.25)$ |
| FGD | $U(0.025, 0.5)$ |

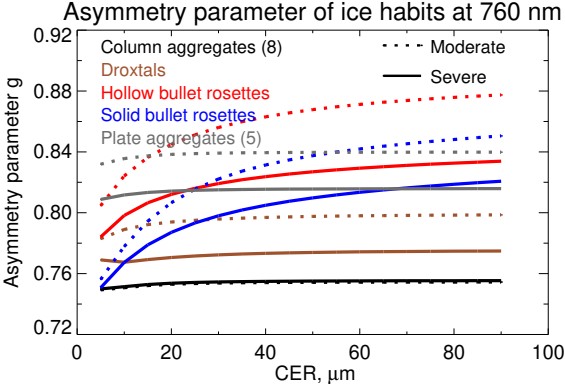

**Figure 7.** Ice crystal asymmetry factor at 760 nm as a function of CER. Different colours indicate difference ice crystal habits; dashed and solid lines refer to moderately and severely roughened crystals, respectively.

scattering phase function (and so TOA reflectance) is weakly dependent on particle size but more strongly dependent on
shape (crystal morphology and aspect ratio) and roughness (van Diedenhoven et al., 2014). In particular, as the asymmetry
parameter $g$ describes the forward vs. backward hemispheric distribution of scattered light, clouds with a similar scaled optical
thickness $\tau_s = \tau(1 - g)$ have a similar scattering signal, where $\tau_s, \tau$ are the scaled and standard COT, respectively. The LUT
interpolation is therefore performed by matching the scaled COT for each simulation's habit and roughness with the equivalent
for the severely-roughened 8-element column aggregates model (Yang et al., 2013a) used as default. This is calculated as
$\tau(1 - g)/(1 - g_{c8e})$ where $g_{c8e}$ is the asymmetry parameter for that default model. Relevant values of $g$ are shown in Figure
7. Key points here are that the column aggregates model has lower $g$ than the others, meaning a higher scaled COT. This habit
model is fairly independent of CER compared to the others, which is a consequence of CER-dependent aspect ratios in other
Yang et al. (2013a) ice crystal models. Increasing crystal roughness in general decreases $g$. Note that this concept of scaled





COT is useful to convert between equivalent cloud optical properties retrieved using different techniques and assumptions (e.g.,
van Diedenhoven et al., 2020).

While the bulk of simulations are performed as perturbations to the cloud vertical structures described in Sections 3.2.5 and 3.2.4, the vertically-homogeneous profile (with added perturbations) is used to provide an alternative for some ensemble members. Near-homogeneous profiles have been observed for liquid (e.g. Wood, 2005; Korolev et al., 2007; Li et al., 2015) and ice (e.g. Feofilov et al., 2015; Krämer et al., 2020) clouds in some circumstances. FGD perturbations are based on Figure 4 as
well as the literature (Thorsen et al., 2011; Zhang and Klein, 2013; Richardson et al., 2019). Aerosol variations were likewise informed by statistics of satellite data (e.g. Sogacheva et al., 2020, although neglecting extreme events).

Surface albedo magnitudes and spectral slopes were based on Figure 6 and the source ECOSTRESS data going into that (Meerdink et al., 2019); they are referred to hereafter as 'ocean', 'land', and 'snow' as shorthand for low, moderate, and high-albedo regimes. The ensemble size means that each combination of cloud phase and surface category contains approximately
15,000 simulations. Surface BRDF effects are not considered for two main reasons. The first is that they are expected to be small, both because OCI and OLCI are single-view sensors and because cloudiness increases the proportion of diffuse light reaching the surface, decreasing the effects of BRDF on the TOA light field. Note that additionally both OCI and OLCI are tilted to decrease the magnitude of Sun glint BRDF features over ocean. The second reason is that libRadtran's DISORT solver run in pseudospherical atmosphere mode does not presently support non-Lambertian surfaces (Buras et al., 2011), meaning
such simulations would have to be run in plane-parallel mode. This is relevant because the plane-parallel assumption introduces systematic biases in the TOA simulation, which become particularly significant at solar zenith angles larger than $60°$ or so, commonly encountered at mid- and high latitudes (see discussion in e.g. Zhai and Hu, 2022). Reverting to plane-parallel simulations to include a BRDF would then introduce a systematic error source.

Several other known sources of uncertainty are not examined in this exercise because they are either computationally in-
tractable, expected to be negligible, or could lead to significantly non-Gaussian perturbations for which the OE formalism breaks down. These include 1D radiative transfer as opposed to 3D; the use of a pseudospherical atmosphere as opposed to full spherical; sensor geometric and channel-to-channel registration; the use of $0.01\,\mathrm{nm}$ spectral spacing for the radiative transfer; the number of angular streams used in the DISORT solver; the use of a scalar rather than vector RT code; the choice of solar reference spectrum; and mixed-phase or multi-layer clouds.

Inspection of $\boldsymbol{y}_{LUT} - \boldsymbol{y}_\delta$ reveals (not shown) that forward model error is generally unbiased and its magnitude is somewhat correlated with the TOA reflectance (more strongly in channels with stronger $O_2$ absorption). Here the forward model uncertainty $\sigma_F$ is estimated by, for each channel, surface type, and cloud phase, ordering and binning the simulated reflectance in each channel from lowest to highest (1,500 simulations per bin) and calculating the 68th percentile (i.e. one standard deviation point) of absolute forward model error ($|\boldsymbol{y}_{LUT,i} - \boldsymbol{y}_{\delta,i}|$) within each bin. Figure 8 shows the results of linear fits to these binned
values for OLCI: in the retrieval, $\sigma_F$ for a given observation is taken as the result of this linear fit, with a floor equal to the root mean square error on the relationships shown in the legend. The spectral forward model uncertainty correlation $r_F$ is estimated for each of the six classes from the Spearman (rank) correlation of $\boldsymbol{y}_{LUT} - \boldsymbol{y}_\delta$ between different wavelengths. Correlations





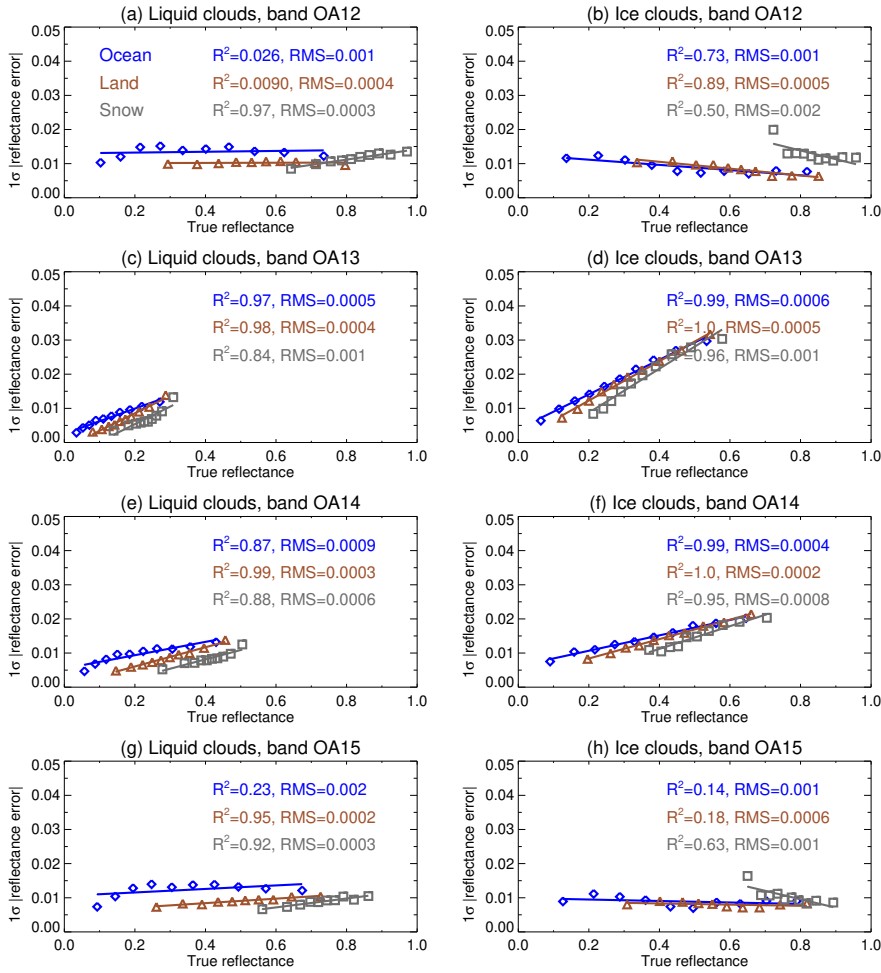

**Figure 8.** Forward model TOA reflectance uncertainty for OLCI. Left and right columns indicate liquid and ice-phase cloud simulations, and rows indicate the four OLCI channels used in CHROMA. Points and lines show the binned data and linear fits, with fit coefficients of determination and root mean square error shown in legends. Blue, brown, and grey show ocean-like, land-like, and snow-like surface albedo respectively.

for OLCI are shown in Figure 9. Spearman correlation is used instead of the more common Pearson due to a small number of extreme (and likely unrepresentative) outliers in the ensemble.

Figures 8 and 9 present results for OLCI rather than OCI because the smaller number of channels allows for a clearer visual presentation. Numerical values and spectral patterns are similar for OCI: for both sensors, channels with similar $O_2$ absorption strength (cf. Figure 1) have similar spectral uncertainty correlation. Correlations decrease for brighter surfaces as these amplify atmospheric absorption effects. However, all correlations are positive and can be almost unity, implying strong degeneracy between the channels in terms of information content. The steepest gradients of uncertainty vs. TOA reflectance tend to be





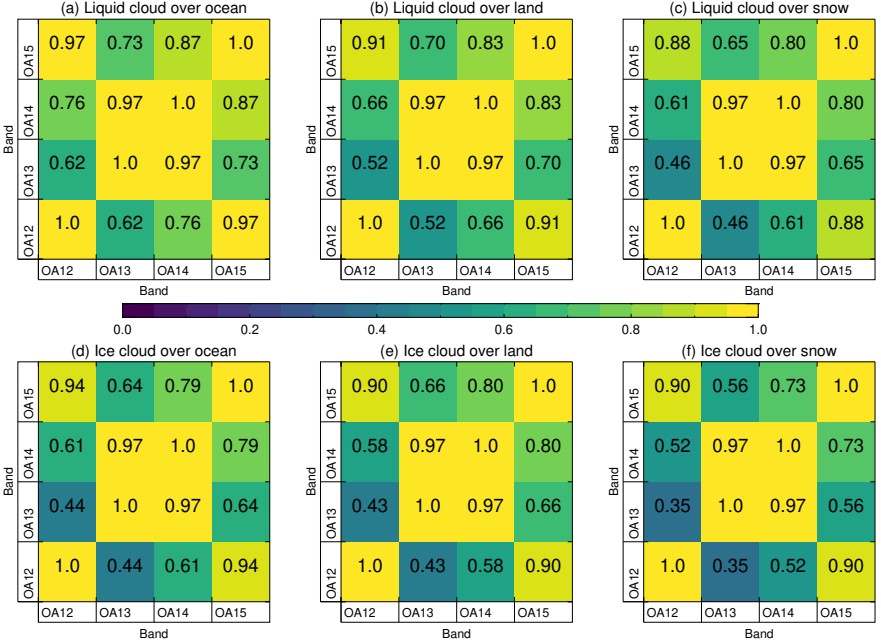

**Figure 9.** Spectral forward model TOA reflectance uncertainty correlation for OLCI. The top row shows results for liquid phase clouds, and the bottom for ice clouds. Columns show, from left to right, results for ocean-like, land-like, and snow-like surface albedo respectively. The colour scale and numbers in each grid box (rounded to two significant digits) are equivalent.

associated with the strongest $O_2$ absorption, which makes intuitive sense as the TOA signal is more strongly dependent on the within-cloud profiles of scattering and absorption in those cases. In weak absorption, and for the brightest surfaces, gradients are weaker and can sometimes be negative: in these cases the TOA signal is most sensitive to cloud-top properties so the effects of cloud profile variations become smaller. Magnitudes of forward model uncertainty vary from around 2-8 % of the TOA signal in most cases, i.e. similar or greater than systematic calibration uncertainty. This implies that improving retrievals

would benefit more from the use of additional channels providing constraints on these forward model assumptions as opposed to calibration improvements.

## 4   Retrievals on synthetic data

### 4.1   Single-layer clouds

#### 4.1.1   Setup

This section presents results of the CHROMA algorithm applied to the ensemble of simulated measurements described in Section 3.3.2 that were used for the forward model uncertainty parameterisation. As this was a fairly simple parameterisation




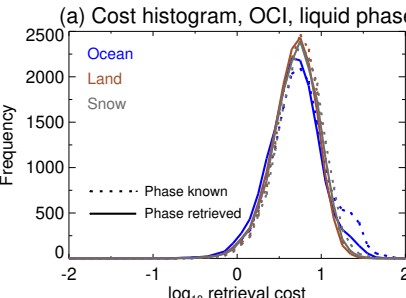
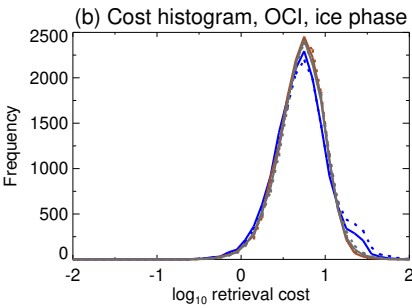

**Figure 10.** Retrieval cost histograms for the simulated single-layer cloud systems, for OCI. Solid and dashed lines show results for best-fit phase vs. known true phase, respectively. Blue, brown, and grey indicate ocean-like, land-like, and snow-like simulations. The left panel shows results for liquid phase simulations, and the right for ice phase simulations.

it provides a test of how generally that formulation holds for the simulated data without the analysis being too circular. In line with this, random and systematic errors are added to the simulated measurements prior to the retrieval. The prior surface albedo is assumed to be unbiased but uncertain, i.e. for a given simulation a random error (Gaussian with standard deviations

0.01 for ocean-like simulations, or 0.05 for land/snow-like simulations) is added to the true (simulated) albedo to generate the (assumed) prior. The retrieval is run once for each of the two phase LUTs (Table 1) and retrieved phase is chosen based on the model returning the lowest retrieval cost. This is essentially how the retrieval would be performed on real satellite observations, in the absence of an externally-available cloud phase mask (although one is expected to be available for OCI, Coddington et al., 2017). Differences between the results of cost-based vs. prior phase knowledge are discussed. The bulk of the analysis focuses

on COT and CTP results for the OCI sensor; OLCI results are generally qualitatively and quantitatively very similar.

### 4.1.2 Retrieval results

Retrieval cost histograms are shown in Figure 10. These are similar for both cloud phases and all three surface types, indicating that the uncertainty model is roughly equally appropriate for all regimes. In theory these distributions should follow a chi-square distribution with degrees of freedom equal to the number of degrees of freedom in the retrieval; in practice this is difficult to

estimate given the strong correlations in forward model uncertainty and the potential for more significant contribution of the surface albedo prior in some cases (i.e. optically-thick clouds) than others. Almost all retrievals converge with cost under 30, suggesting that in real-world applications a cost above this could be used to indicate a drastic problem with the retrieval as a quality check. This will be assessed in a follow-up analysis using real OLCI data. Histograms for the case of phase known vs. determined by best-fit are also fairly similar, indicating that either the correct phase is commonly chosen or that, when

the incorrect phase is chosen, the misfit is not too large. The one exception is for clouds over water surfaces, where a sub-population of cloud retrievals are better fit with the wrong model than the correct one. Examination (not shown) reveals these to be a subset of simulations where the cloud vertical profile was strongly different from that assumed in the LUT. Results are similar for OLCI (not shown), except due to having half the number of channels and lower overlap the distribution is broader





and shifted slightly left. For both phases **A** for surface albedo is close to 0 for all COTs over water, indicating retrieved albedo

is determined by the prior. For land-like and snow-like simulations elements of **A** are more variable but tend to decrease rapidly

as COT increases past $\sim 3$. This is expected as, for an opaque cloud, the TOA signal is weakly sensitive to the surface albedo.

Retrieval errors binned as a function of COT are shown in Figure 11. As COT increases, as expected the error on COT,

CTH, and CTP tends to decrease while that on surface albedo increases up to around the prior uncertainty. The retrievals are,

in most cases, unbiased - although for cases over snow with the highest albedo, in some cases a low-biased albedo is retrieved,

compensated by optically-thinner clouds retrieved at a higher altitude (lower pressure) than is true. The bottom row shows that

retrieval cost has little skill (slightly better than chance) at picking the correct phase for liquid clouds, while for ice clouds it

allows phase to be determined approximately 80-90 % of the time. The phase dependence is in fact related to the cloud altitude

ranges covered by the simulations and LUT: as the clouds are essentially gray scatterers across this wavelength range, the

primary sensitivity is to brightness (related to the combination of COT and phase function) and CTP, rather than COT directly.

This points to the utility of extra OCI channels, such as the SWIR, for OCI phase determination for an additional constraint on

the retrieval (Coddington et al., 2017). Note OLCI lacks SWIR channels.

The CHROMA algorithm retrieves two quantities for which the PACE mission has performance goals: COT and CTP. These

goals are defined such that, on global average, 65 % of retrievals should have errors of a given size or lower. These goals

are 25 % for liquid-phase COT, 35 % for ice-phase COT, and 60 mb for CTP. The CTP goal is only required for clouds with

COT$\geq 3$, but it is informative to examine across the range. Attainment of these goals is shown in Figure 12. This shows that,

again, retrieval quality increases with increasing COT. Knowledge of the correct phase improves compliance with the COT

goal, but had little effect on CTP-consistent with the previous point that retrieved COT is more sensitive to the cloud model

assumed while CTP is to the spectral signature of absorption and the cloud reflectance. Attainment of these goals is easier for

darker surfaces than brighter, due to the increased contrast between the surface and cloud in the window channel. This implies

that stronger prior constraints on surface albedo, which makes this separation easier, may be helpful. Results are similar for

OLCI (not shown). Goals are more readily achieved for ice clouds than liquid-phase clouds (although recall that the two phases

have different altitude distributions in the simulations, which may be important). It is also worth noting that the bispectral

approach for joint COT/CER retrieval, which has been applied to MODIS and other sensors (Platnick et al., 2003, 2021), will

also be applied (using a MODIS-like retrieval approach) to OCI - providing a separate COT data source.

Table 4 summarises numerically the CTP retrieval performance for both sensors. Patterns as a function of cloud phase and

surface type are similar between OCI and OLCI. Statistical metrics for OLCI tend to be up to 5-10 % better than for OCI. This

is because OCI's broader, overlapping channels mean the $O_2$ absorption signature is less distinct than for OLCI (Figure 1).

However, the broad similarity in statistics for the two sensors, combined with OLCI's similar spatial resolution, confirms the

utility of this sensor as a proxy for testing the CHROMA algorithm prior to PACE launch. Table 4 summarises statistics for the

simulations where PACE has CTP performance goals (COT$\geq 3$); if this is relaxed to consider COT$\geq 0$ (not shown) statistics

degrade slightly although the same category-to-category and sensor-to-sensor patterns hold.

The final component of this part of the analysis concerns two key assumptions in the retrieval. The first is cloud vertical

extent (via FGD); retrieval errors binned as a function of this are shown in Figure 13. Except for retrievals over snow, COT





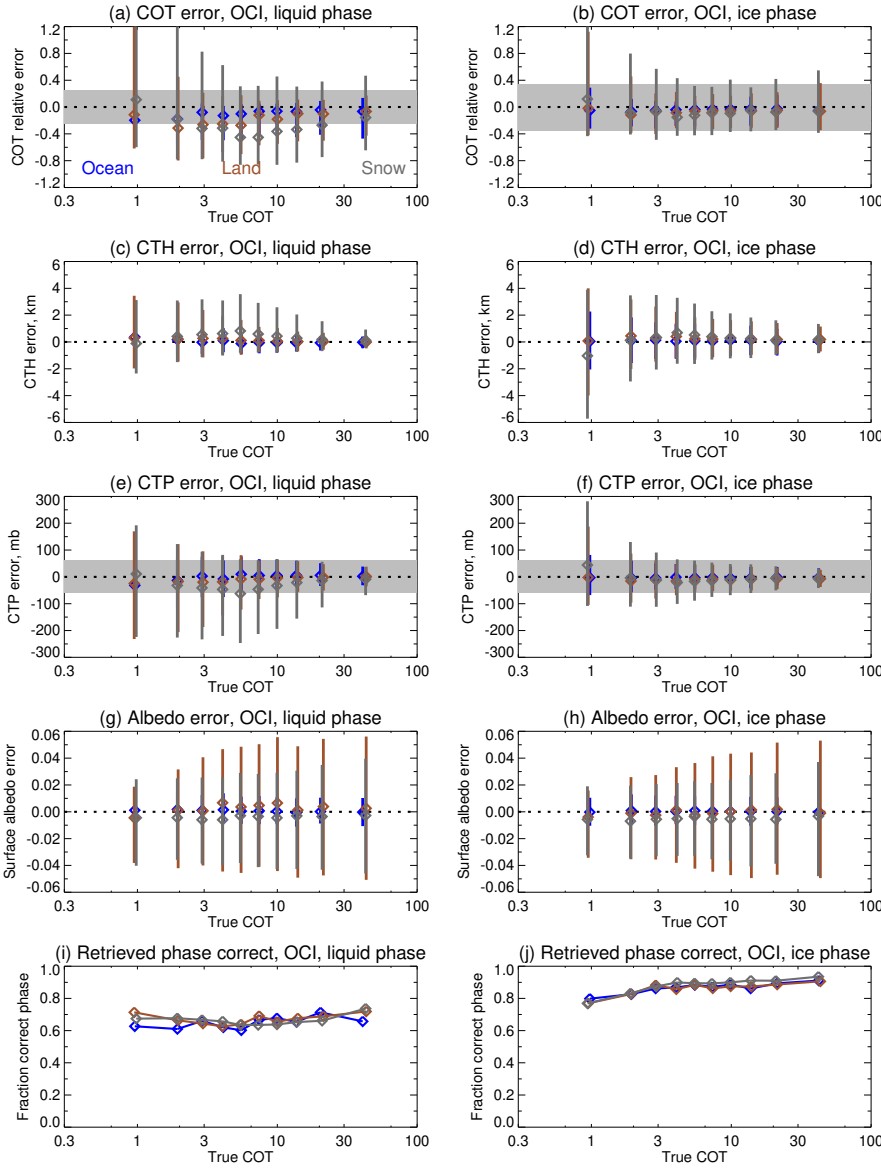

**Figure 11.** Retrieval errors binned as a function of COT for the simulated single-layer cloud systems, for OCI. Points indicate the median retrieval error and vertical lines the central $1\sigma$ of results within each bin. Blue, brown, and grey indicate ocean-like, land-like, and snow-like simulations. The left column shows results for liquid phase simulations, and the right for ice phase simulations. From top downwards, rows indicate COT, CTH, CTP, and surface albedo. The very bottom row shows the fraction of data where the best-fit phase (liquid or ice) is the correct phase for the simulation. Shaded grey indicates the PACE mission goal uncertainties: 25 % for liquid-phase COT, 35 % for ice-phase COT, and 60 mb for CTP (the latter for COT$\geq$ 3 only). The dotted line indicates an error of zero.





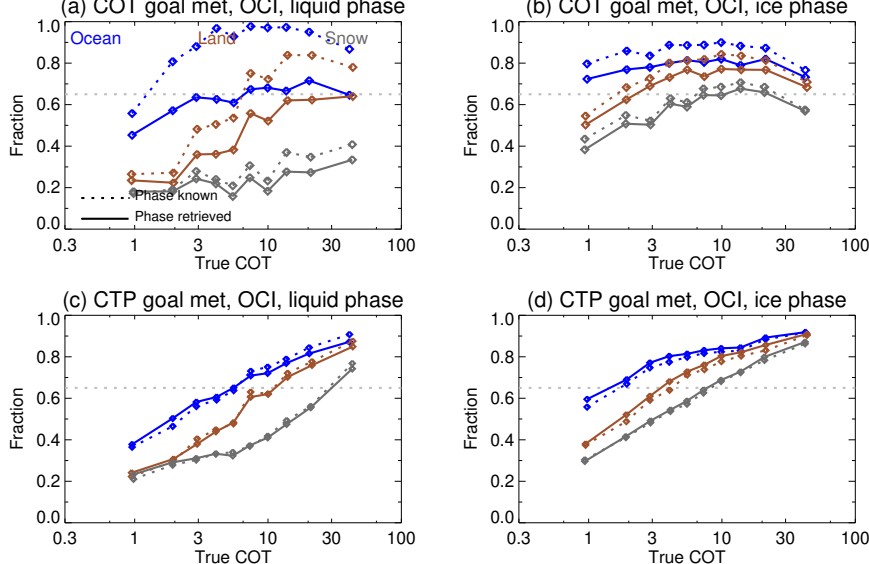

**Figure 12.** Fraction of points where the retrieval error meets the PACE mission goal uncertainties (25 % for liquid-phase COT, 35 % for ice-phase COT, and 60 mb for CTP for COT$\geq$ 3) as a function of COT, for OCI. The dotted pale grey line shows the 65 % threshold for achieving these goal uncertainties. Solid and dashed lines show results for best-fit phase vs. known true phase, respectively. Blue, brown, and grey indicate ocean-like, land-like, and snow-like simulations. The left panel shows results for liquid phase simulations, and the right for ice phase simulations.

**Table 4.** Statistics of CTP retrieval error on simulated single-layer cloud scenes, for OCI and OLCI, for the subset of cases where COT$\geq$ 3. Results for the case where cloud phase is known beforehand. MAE and RMSE are median absolute error and root mean squared error.

| Surface | Rank correlation | | Median bias, mb | | MAE, mb | | RMSE, mb | | Fraction within goal | | Fraction within uncertainty | |
|---|---|---|---|---|---|---|---|---|---|---|---|---|
| | OCI | OLCI | OCI | OLCI | OCI | OLCI | OCI | OLCI | OCI | OLCI | OCI | OLCI |
| | | | | | | Liquid phase clouds | | | | | | |
| Ocean | 0.92 | 0.94 | 1 | 4 | 31 | 30 | 57 | 53 | 0.75 | 0.77 | 0.61 | 0.65 |
| Land | 0.84 | 0.86 | -5 | -2 | 41 | 37 | 91 | 81 | 0.64 | 0.68 | 0.66 | 0.70 |
| Snow | 0.64 | 0.69 | -16 | 12 | 67 | 60 | 140 | 130 | 0.45 | 0.50 | 0.73 | 0.73 |
| | | | | | | Ice phase clouds | | | | | | |
| Ocean | 0.95 | 0.95 | -3 | -2 | 27 | 26 | 46 | 43 | 0.83 | 0.85 | 0.62 | 0.69 |
| Land | 0.91 | 0.92 | -9 | -7 | 30 | 30 | 62 | 56 | 0.77 | 0.80 | 0.67 | 0.73 |
| Snow | 0.83 | 0.83 | -12 | -11 | 38 | 36 | 82 | 78 | 0.67 | 0.71 | 0.68 | 0.73 |

error is insensitive to this, which is expected as information on this parameter comes primarily from the absorption-free window
channel. For CTP, however, there is a clear trend where CTP is progressively underestimated (i.e. retrieved cloud is too high) if





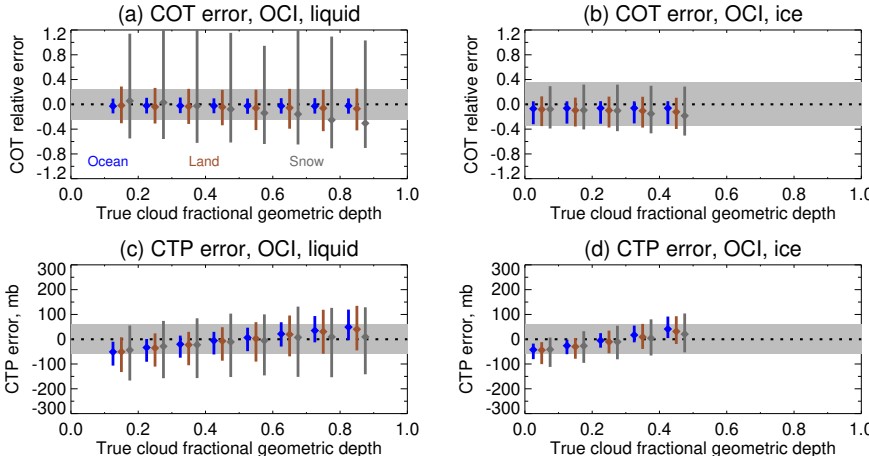

**Figure 13.** Retrieval error as a function of cloud FGD, for OCI, for the case where cloud phase is known beforehand. Points indicate the median retrieval error and vertical lines the central $1\sigma$ of results within each bin. Blue, brown, and grey indicate ocean-like, land-like, and snow-like simulations, slightly offset from one another horizontally for clarity. The top row shows COT and the bottom CTP. The left column shows results for liquid phase simulations, and the right for ice phase simulations. Shaded grey indicates the PACE mission goal uncertainties: 25 % for liquid-phase COT, 35 % for ice-phase COT, and 60 mb for CTP (the latter for COT$\geq$ 3 only). The dotted line indicates an error of zero.

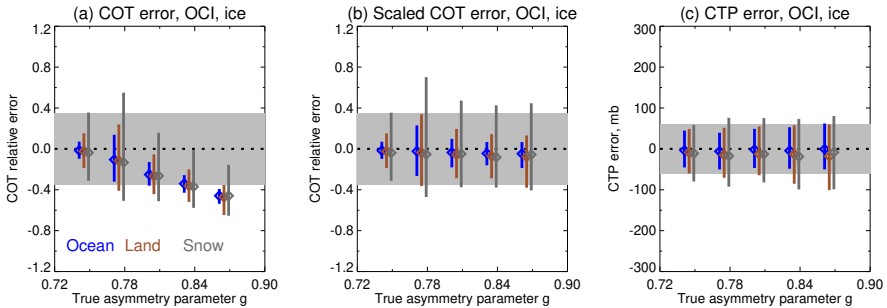

**Figure 14.** Retrieval error as a function of ice crystal asymmetry parameter, for OCI, for the case where cloud phase is known beforehand. Blue, brown, and grey indicate ocean-like, land-like, and snow-like simulations, slightly offset from one another horizontally for clarity. From left to right, panels show COT, asymmetry-corrected COT (Section 3.3.2), and CTP. Shaded grey indicates the PACE mission goal uncertainties: 35 % for ice-phase COT and 60 mb for CTP (the latter for COT$\geq$ 3 only). The dotted line indicates an error of zero.

clouds are shallower than assumed (FGD is lower) and overestimated in the opposite case. The change in median error across the range of FGD can approach the $\pm 60\,\mathrm{mb}$ total error goal. Future work will therefore focus on ways to add OCI's other spectral channels to the retrieval and allow it more flexibility on retrieving parameter(s) relevant to within-cloud scattering. Unfortunately for OLCI, Fischer and Preusker (2021) showed that A-band capabilities alone at these spectral resolutions are often insufficient for this.






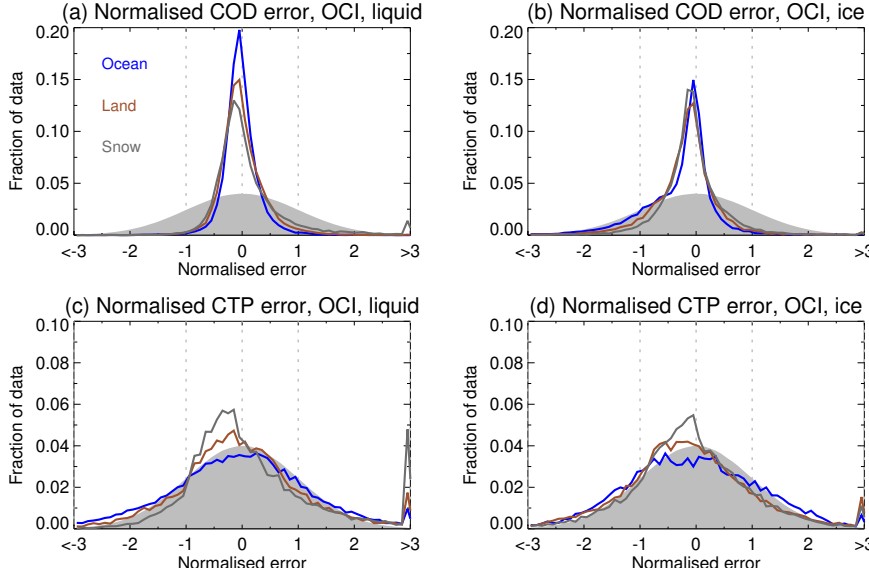

**Figure 15.** Histograms of normalised retrieval error for OCI, for the case where cloud phase is known beforehand. Blue, brown, and grey indicate ocean-like, land-like, and snow-like simulations. The top row shows COT and the bottom CTP. The left column shows results for liquid phase simulations, and the right for ice phase simulations. Shaded grey shows the theoretical Gaussian distribution which would be observed if uncertainty estimates were calibrated perfectly. Vertical dotted lines indicate normalised errors of -1, 0, and +1.

The second assumption is the ice crystal shape and roughness which, as previously discussed, influence the relationship between COT and TOA reflectance as a function of its asymmetry parameter $g$ (Section 3.3.2, Figure 7). Retrieval errors as a function of $g$, for ice cloud simulations, are shown in Figure 14. In contrast to the FGD assumption, here it is CTP retrieval which is fairly invariant to the assumption. As $g$ increases beyond the assumed value (for severely roughened column aggregates) of ∼0.75, retrieved COT becomes increasingly low-biased. The asymmetry-corrected COT $\tau(1-g)/(1-g_{c8e})$, however, becomes almost unbiased and of similar error across the range of true $g$. This is consistent with the results of van Diedenhoven et al. (2020) and implies that the combination of OCI with PACE's polarimeters will be able to provide estimates of COT corrected for CHROMA's assumed ice model in those situations where the asymmetry parameter is higher (Figure 7).

### 4.1.3 Uncertainty estimates

Beyond evaluating the retrievals, it is useful to evaluate the retrieval-level uncertainty estimates provided by the OE technique. Several methods can be used for this, though a key point (Sayer et al., 2020) is that the predicted uncertainty represents an estimate of the precision of the retrieved quantity (as a mean and standard deviation of a Gaussian probability distribution), while an error (i.e. retrieved value minus true value) is a discrete number. Essentially, the former is an estimate of dispersion and the latter a draw from a distribution of possible errors. The goal is to assess whether the statistical distribution of observed errors is consistent with that expected based on the OE-provided uncertainties.



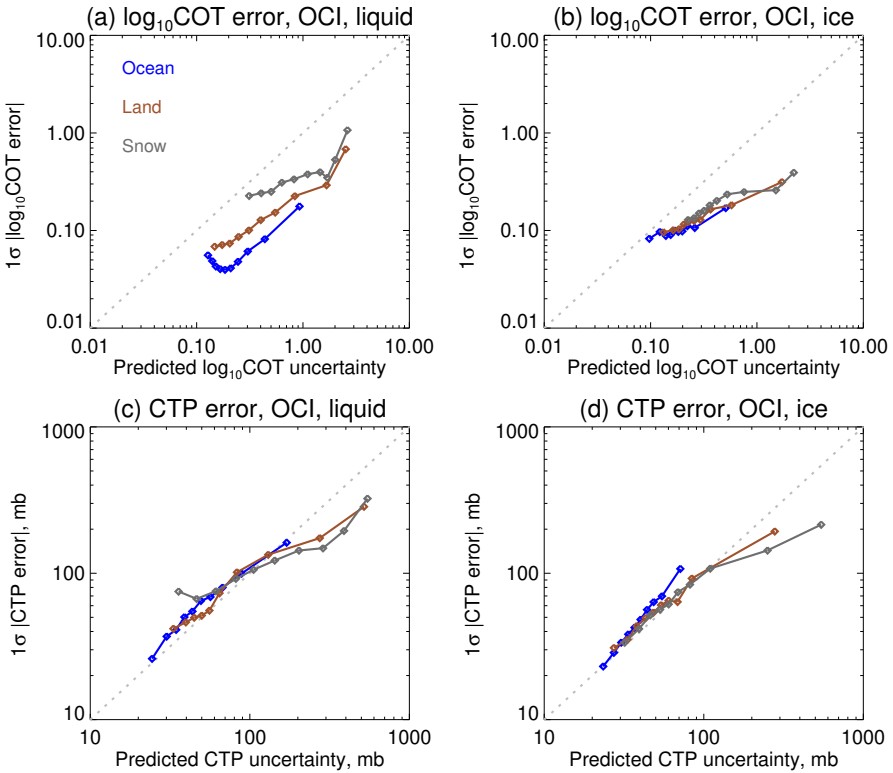

**Figure 16.** $1\sigma$ points of absolute retrieval error binned as a function of OE-provided retrieval-level uncertainty estimate, for the case where cloud phase is known beforehand. Blue, brown, and grey indicate ocean-like, land-like, and snow-like simulations. The top row shows COT and the bottom CTP. The left column shows results for liquid phase simulations, and the right for ice phase simulations. The 1:1 line (representing perfect agreement with theory) is dotted.

This analysis follows Sayer et al. (2020) and evaluates retrieval uncertainties in two ways. The first (Figure 15) is using histograms of normalised error, i.e. retrieval error divided by that retrieval's estimate of retrieval uncertainty. If the retrieval is unbiased and its uncertainty budget is well-characterised, then these histograms should approach Gaussians with mean 0 and standard deviation of 1. Using this measure, Figure 15 shows that the COT retrieval is underconfident (i.e. retrieval errors are
smaller than predicted) as the distribution is somewhat narrower than expected (for all combinations of cloud phase and surface type). Uncertainties for CTP, however, are much more in line with expectations. This may imply that the spectral pattern of uncertainty is captured well by the model (as spectral shape drives the CTP retrieval), but the absolute magnitude in the window channels (which drives the COT/albedo separation) is overestimated on average.

Figure 16 provides an assessment of whether the retrieval is skilful at telling low-uncertainty situations from high-uncertainty
ones. This is achieved by sorting the simulations by retrieval uncertainty, and dividing into 10 bins. The median of the predicted uncertainty in each bin is then compared to the 68th percentile (i.e. $1\sigma$ point of a Gaussian distribution) of absolute retrieval error. If the uncertainties are well-calibrated, the points should line up along the 1:1 line. For COT, there is some skill at telling





**Table 5.** Sampling of geophysical parameters for multi-later (ice above liquid) cloud simulations. Quantities not given here are as in Tables 2 and 3.

| Quantity | Sampling distribution |
|---|---|
| | Lower liquid cloud layer |
| CTH (km) | $U(0.5, 6)$ |
| | Upper ice cloud cloud layer |
| $\log_{10}$(COT) | $N(-0.2, 0.5)$ truncated at -1.5 and 2 |
| CTH (km) | $U(8, 16)$ |
| FGD | $U(0.025, 0.25)$ |

low- from high- uncertainty situations although (consistent with Figure 15) the actual errors are factors of 2-5 smaller than expected. For CTP, the relationship is much closer to 1:1, with a tendency to slightly underestimate uncertainty on the low end

and slightly overestimate uncertainty on the high end.

The patterns hold for both phases (though ice COT uncertainty is better-calibrated than liquid cloud), all three surface types, and (not shown) both sensors and cases of known vs. retrieved cloud phase. These results provide some confidence that the uncertainty estimates provided by the retrieval will be useful to data users who wish to filter or screen the data, or incorporate uncertainty estimates into downstream applications (such as data assimilation or incorporation as priors into other retrievals).

**4.2 Multi-layer clouds**

Historically and at present many cloud altitude retrieval algorithms for passive satellite sensors (like CHROMA) assume a single-layered cloud system. Multi-layer cloud systems, however, are not uncommon, especially around the Equator and storm tracks in both hemispheres (Subrahmanyam and Kumar, 2017; Marchant et al., 2020). In such cases, these algorithms typically report a height intermediate between that of the multiple layer(s) present, unless the uppermost layer is optically thick (Naud

et al., 2007; Sayer et al., 2011). It is therefore useful to briefly examine the expected behaviour of CHROMA in these situations. One common case (although other structures exist), examined here, is for an optically-thin ice cloud (cirrus) above a liquid cloud. For this purpose a further synthetic ensemble of 3000 simulations has been created, modifying the previous single-layer simulations as indicated in Table 5. The CHROMA algorithm is then, as before, rerun in the standard single-layer configuration (adding on sensor radiometric error and albedo prior error).

Figure 17 summarises some main results for data divided into 8 COT bins. In these cases, the ideal 'truth' values that would be retrieved correspond to the total (liquid plus ice) COT, and the CTP of the upper (ice phase) cloud. Overall, total COT is underestimated as ice COT increases - in part because the asymmetry parameter of liquid water droplets is higher than that for most ice crystal habits, yielding retrieval behaviour consistent with Figure 14(a). The fraction of retrievals retrieved as ice increases roughly linearly with $\log_{10}$(ice COT) although, as shown in Figure 11(i,j), for single-layer cases misidentification of

liquid-phase clouds as ice-phase clouds was more common than the converse.





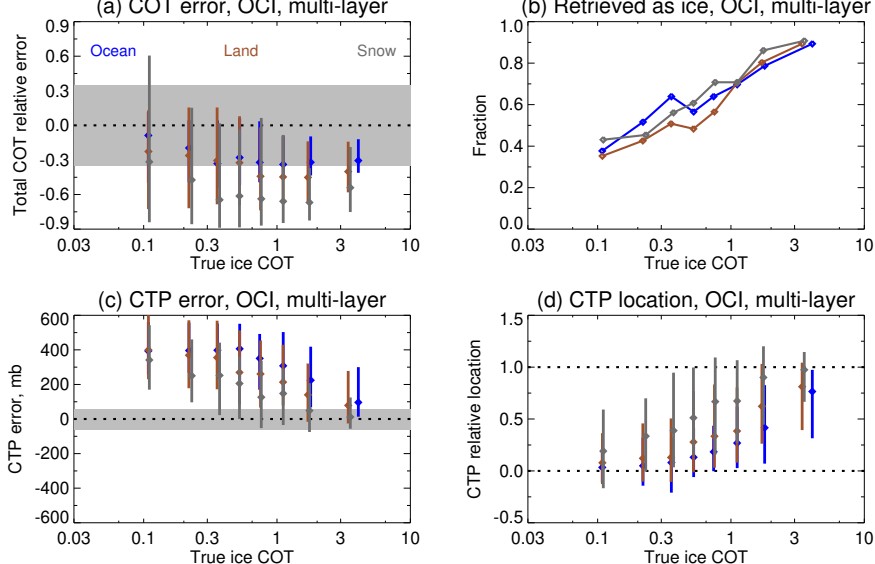

**Figure 17.** Retrieval performance binned as a function of ice COT for the simulated multi-layer cloud systems, for OCI. Blue, brown, and grey indicate ocean-like, land-like, and snow-like simulations. The left column shows retrieval errors for (a) total (liquid plus ice) COT and (c) CTP. In these panels points indicate the median retrieval error and vertical lines the central $1\sigma$ of results within each bin, the dotted line an error of zero, and shaded grey indicates the PACE mission goal uncertainties: 35 % for ice-phase COT, and 60 mb for CTP (the latter for COT $\geq 3$ only). Panel (b) shows the fraction of data where the best-fit cloud optical model was ice. Panel (d) shows the median (points) and central $1\sigma$ (lines) of the retrieved CTP's location relative to the lower and upper cloud layer CTPs (dotted lines at 0 and 1, respectively).

Retrieved CTP for the two-layer simulations is severely overestimated, due to the optical dominance of the lower liquid layer (which is intentionally shifted to higher COTs relative to the upper layer for these simulations; Table 5). Figure 17 expresses this in two ways. Figure 17(c) shows retrieved CTP error (i.e. retrieved minus upper ice layer CTP) as a function of ice COT, equivalent to Figure 11(f). The median error for the lowest ice COT bin corresponds to roughly the median CTP difference

between upper and lower simulated cloud layers. Figure 17(d) demonstrates this relative positioning more directly: here a relative location of 0 indicates a cloud retrieved at the top of the liquid layer, and 1 that the cloud was retrieved at the top of the ice layer. For ice COT around 0.1 the ice layer is effectively not seen by the algorithm, and the retrieved pressure is close to that of the lower (liquid phase) layer. An ice COT of around 3 seems to be sufficient to largely 'shield' the TOA signal from the lower layer, and the retrieved CTP is closer to the upper (ice phase) layer. For intermediate ice COT, the retrieved CTP tends

to lie in between that of the two cloud layers.

Retrieval cost histograms increase by roughly a factor of 2-3 compared to single-layer cases (not shown), peaking around 10. As this lies within the range of cost for single-layer cases (Figure 10) this implies that cost alone may not be an effective way to flag multi-layer cases. Normalised uncertainty estimates also lack skill (not shown) for these cases, due to the large non-Gaussian behaviour of the errors (i.e. large biases), which is expected for situations where the underlying retrieval forward





model is inappropriate for the situation at hand (Povey and Grainger, 2015). The results in this section hold for all surface types, and for OLCI (not shown). While these results are specific to the simulation's parameters only, more generally it therefore seems likely that the retrieval will encounter similar biases to single-layer cloud retrieval algorithms applied to other sensors.

## 5   Conclusions

This study presents a new algorithm, CHROMA, whose primary purpose is retrieval of cloud top altitude. The algorithm uses
multispectral measurements across the $O_2$ A-band so is sensitive to cloud-top pressure (as $O_2$ is a well-mixed gas), although this can be converted to cloud top height or temperature using meteorological profiles. The motivation for algorithm development is the OCI sensor on the forthcoming NASA PACE satellite mission (https://pace.gsfc.nasa.gov), although it is applicable to other spaceborne sensors with similar spectral and spatial characteristics such as OLCI. As such, a follow-up study will present a broader-scale application of the CHROMA algorithm applied to on-orbit OLCI measurements.

As with other passive cloud altitude remote sensing techniques, the primary assumption affecting the accuracy of the retrieved altitude is the assumed vertical structure of the cloud, in terms of its extent, extinction profile, and number of distinct cloud layers. Unfortunately these parameters are not well-constrained by the available satellite measurements. Sensitivity of retrieved CTP to assumed cloud particle makeup (various sizes of water droplets or ice crystals of different shapes/roughness) is comparatively minor, but more strongly affects the COT also retrieved by the technique. Surface albedo is additionally
retrieved but with a strong prior constraint (which tends to be dominant, expect for optically-thin clouds).

CHROMA is a Bayesian retrieval using the Optimal Estimation framework, which provides retrieval-level uncertainty estimates for each parameter. Retrieval simulations suggest that these uncertainty estimates are skilful and quantitatively reasonable for CTP, although underconfident (overestimate error) for COT, and that the OCI sensor and algorithm should be able to meet the PACE mission goal for CTP error of $\pm60$ mb for 65 % of opaque (COT$\geq 3$) single-layer clouds on global average. The
aforementioned follow-up study using OLCI data will assess how well the performance of simulated retrievals translates to real data.

The algorithm has several synergies with the two multi-angle polarimeters, HARP2 (Martins et al., 2018) and SPEXone (van Amerongen et al., 2019), which will also fly on the PACE mission. Firstly, the retrieved CTH could be used to coregister the different view angles from these sensors to cloud altitude as opposed to surface altitude when retrieving cloud (or aerosol
plume) properties from the polarimeters. Several cloud properties will be retrievable from them (solo or combined with OCI) that are unavailable from OCI alone. For example, HARP2's hyperangular sampling will enable retrievals of liquid cloud effective radius and variance from polarised cloud bow features (Alexandrov et al., 2012). It is also possible that HARP2's multiple views could be used for a parallax-based CTH retrieval (Moroney et al., 2012), which would allow for independent comparison with OCI results, although as its spatial resolution is likely to be ∼2.5-4 km dependent on view angle this may not
be precise enough to be useful. CHROMA could also be applied directly to the SPEXone sensor, as this has similar spectral characteristics in the $O_2$ A-band to OCI; its five view angles can provide additional constraints on cloud geometric thickness (Ferlay et al., 2010). Two disadvantages of SPEXone, however, are its comparatively narrow swath (100 km) and coarser





pixel footprint (∼5 km), meaning a greater proportion of cloudy pixels will sample broken cloud scenes (although focused on cloud-free scenes cf. sampling results in Krijger et al., 2007; Remer et al., 2012), which present their own difficulties.

The combination of the different sensors also presents opportunities for pixel-level correction of imager cloud retrievals using polarimeter-derived properties (e.g. ice asymmetry parameter), as has been done for POLDER and MODIS (van Diedenhoven et al., 2020). Further, the differences between total and polarised reflectances could be indicative of ice clouds over water, providing an external multi-layer flag for the OCI retrievals.

In the future, several enhancements could make the algorithm more capable at the expense of breaking compatibility with

OLCI (and requiring significant development time). The first would be to combine the retrieval with the MODIS/VIIRS heritage optical properties (COT/CER) approach, rather than have the two as separate retrieval algorithms. This is expected to lead to a modest improvement as the additional channels would provide some extra constraint on COT, and having cloud-top CER retrieved would decrease one uncertainty source for the CTP retrieval. However this is not the primary uncertainty source, and the resulting large LUT dimensionality would likely mean that a neural network would have to be trained to replace the LUT

in order for data processing to be computationally tractable.

The second approach would be to include cloud vertical extent as an additional retrieved parameter, which (if constrained appropriately) should improve CTP retrieval as well as providing useful scientific information of its own. The simple way to do this would be to add another LUT dimension (which significantly increases computational overhead) and assume a prior value with some uncertainty. Cloud top height and centre of gravity are retrieved by OLCI A-band measurements by

Preusker and Fischer (2021), although as noted by Fischer and Preusker (2021) this additional parameter is not robust in most circumstances. Possibilities for extra OCI channels to improve constraints include the $O_2$ B-band which, while weaker, has a signal across several OCI channels (Figure 3), and NIR/SWIR channels affected by water vapour absorption. These, however, bring additional (surmountable) complications: a need for surface albedo characterisation at those wavelengths and knowledge of the vertical structure of water vapour.

These challenges should be surmountable, and a strength of Bayesian approaches is the ability to account for the information content offered by additional channels in a robust way, provided associated measurement and forward model uncertainties can be quantified. Even without enhancements, however, the presented algorithm should meet the PACE mission's needs and is readily implementable for application to simulated and proxy (e.g. OLCI) data in advance of mission launch.

*Code availability.* The libRadtran radiative transfer package used for radiative transfer calculations in this work and associated files (e.g.

modules for liquid and ice phase clouds) can be obtained from http://www.libradtran.org/doku.php. The MT-CKD continuum absorption code is available at https://github.com/AER-RC/MT_CKD.

*Data availability.* Retrievals based on simulated and eventually real PACE OCI measurements will be freely available from https://oceancolor.gsfc.nasa.gov. Expected OCI RSRs are available from https://oceancolor.gsfc.nasa.gov/data/pace/characterization, and pre-launch OLCI



RSRs from https://dragon3.esa.int/web/sentinel/technical-guides/sentinel-3-olci/olci-instrument/spectral-response-function-data. The refer-ence solar spectrum used can be downloaded from https://lasp.colorado.edu/lisird/data/tsis1_hsrs. The HITRAN absorption data base is avail-able from https://hitran.org. ARM ARSCL data are available from https://adc.arm.gov/discovery/#/results/instrument_class_code::kazrarscl.

*Author contributions.* AMS led development of the algorithm, performed the calculations, developed processing codes, and led preparation of the manuscript. LL, SK, and AI assisted with benchmarking and verification of the radiative transfer. SK and LL helped with historical research and (SK) translation of Russian texts. BvD provided data and advice for ice cloud profiling. All authors provided advice during algorithm development and evaluation, and contributed to drafting and review of the manuscript.

*Competing interests.* The lead author is an Associate Editor for Atmospheric Measurement Techniques.

*Acknowledgements.* LL was funded by the Alexander von Humboldt foundation via the Feodor-Lynen fellowship 2020. BvD was funded by SRON. All other authors were funded by the NASA PACE project. The authors are grateful to J. Elsey (University of Reading, UK), I. Gordon (Harvard University, USA), V. Natraj (JPL, USA), and K. P. Shine (University of Reading, UK) for useful discussions on modeling the $O_2$ A-band and continuum absorption. AMS thanks A. B. Davis (JPL) for interesting historical discussion on $O_2$ remote sensing, and J. M. Gales (SAIC) for assistance with cluster computing. The developers of the libRadtran software, HITRAN data base, and MT_CKD continuum model are deeply appreciated for these powerful, freely-available tools.



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
