# Peer review of "The CHROMA cloud top pressure retrieval algorithm for the Plankton, Aerosol, Cloud, ocean Ecosytem (PACE) satellite mission"

_Atmospheric Measurement Techniques, 2022_

## Author Comment (AC1)

We thank the Reviewers for their reading of and comments on our work. Below, reviewer comments are in **bold** and our responses in regular type.

In addition to changes made in response to reviewer comments, we have expanded one paragraph of the Introduction to provide additional context on top pressure vs. optical centroid (and the benefits of each) as well as details about the importance of spectral sampling and differential absorption strengths.

**Reviewer 1**

**Sayer et al. presented the CHROMA algorithm and provided simulated retrievals for OCI on the forthcoming PACE mission and for OLCI on Sentinel-3. The CHROMA algorithm retrieves cloud optical thickness and cloud top pressure/cloud height, and surface albedo simultaneously and provides uncertainty estimates for each pixel. Based on the simulated retrievals, the authors found that the CHROMA algorithm should meet the PACE mission requirement for CTP.**

**There are new features in the CHROMA algorithm, although the cloud and aerosol retrievals from $O_2$ A band is well-understood. This paper provides important insight about the sensitivity of CTP to the cloud phase and ice cloud habits (asymmetry parameter). The CHROMA algorithm uses cloud top effective radius as a function of cloud top pressure for ice and water clouds separately instead of a homogeneous cloud layer. The cloud layer has a fractional geometric thickness, 0.5 for water clouds and 0.25 for ice clouds. The estimation of the covariance matrix $S_y$ is another relevant feature of this paper.**

**I think CHROMA algorithm has taken into account lots of details in the cloud setup and retrieval. The authors explained the theoretical basis and performance of the algorithm clearly. It is a pleasure to read this paper. I think the paper can be published as is.**

We are grateful for the reviewer's positive opinion of our paper and their recommendation for publication.

**Specific comments:**

**1) Page 9, line 233 'The cloud layer is divided into 5 contiguous sublayers, running from the CTH to the cloud base height (CBH).'**

**It is not clear how the sublayers are defined. Are the top and base heights of the sublayers equally distributed in the cloud layer? Is 5 sublayers accurate enough for the libRadtran simulations for geometric thick clouds?**

The sublayers are of equal geometric height; this has been clarified in the revised text. We also ran a number of simulations using 10 sublayers and found minimally different results in most cases (much smaller differences than contributions to the uncertainty budgets from individual microphysical/macrophysical assumptions.)

We also note (line 235 of the original manuscript) that if a cloud sublayer straddles multiple atmospheric profile layers (of which there are 19), these are split to create additional layers in the atmospheric profile. This ensures that the basic vertical structure of molecular, aerosol, and cloud scattering and absorption is not distorted by the different vertical specification of these differing components.

**2) Page 18, close to the last line '…, which is a consequence of CER -dependent aspect ratio in other Yang et al. (2013a) '**

**Please check if 'other' is correct in the sentence.**

In the revision we have rephrased this to "in other ice crystal models found in Yang et al. (2013a)" to be clearer.

**3) Figures 8 and 9. Caption. Please provide the central wavelengths from OA12 to OA15 in the caption. I think only people work on OLCI would know the band numbers.**

Good suggestion – we'd originally given band names as central band wavelengths (and, indeed, band widths) for OLCI change as a function of across-track position. We have added nominal central wavelengths for the four relevant bands to the captions of these figures.

**4) Figure 9 might need more explanations. In the paragraph started from line 470 in page 20, the first two sentences related to OCI would be better to move to the previous paragraph. This paragraph explained Figure 9 but it was not so straight forward.**

We have reorganized these last two paragraphs as suggested as we agree with the reviewer that the original ordering interrupted the flow of the text a bit.

**Reviewer 2**

**General comments:**

**The manuscript presents the theoretical basis, performance, and usage of an algorithm, called CHROMA, whose main purpose is the retrieval of cloud top pressure or altitude from future PACE OCI measurements. Radiative transfer simulations are extensively used to understand sensitivities and expected performance of CHROMA applied to OCI, also to OLCI measurements. While OLCI measurements are already effective from Sentinel 3A and 3B satellites since 2016, CHROMA applied to OLCI does not actually use real measurements, but only RT of it, which can be surprizing at the first sight. The authors announce it for a follow-up study. The results of this study is that the performance of CHROMA applied to OCI or OLCI should meet the PACE mission goal for CTP error.**

**The abstract of the paper is clear. The paper is well written and easily readable. The authors are generous in providing informations that help to follow the information content of the measurements, the performances and weaknesses of the algorithm. A significative section is dedicated to the estimate of the covariance matrix Sy. Authors don't neglect the fact that cloudy atmosphere are not rarely multilayered, that challenges all cloud algorithm, and in particular the CHROMA algorithm. The given perspectives of the paper are particularly interesting.**

**I think that this manuscript needs only minor revisions before being accepted in AMT. The minor revisions consist in the following: shorten the paper by making the choice to move some figures and tables to an appendix. Indeed, 17 figures and 5 tables are a lot, and a global feeling reading the manuscript is that the figures are not fully exploited and not entirely necessary in the main text. For example Figure 10, Table 4 could be moved to an appendix.**

We're grateful that the reviewer appreciates the content and general style of our manuscript. Although OLCI has been on orbit for a while we stuck to simulations only in order to be able to keep the paper focused on the theory and OLCI-OCI comparisons. We have a second manuscript in preparation showing a global application to OLCI and comparison with other satellite and ground-based data sets. We felt that showing examples of real OLCI data in here would increase the length too much to warrant the additional interest – and in any case hope that a preprint of our second paper will be available in the not too distant future. So the purpose of this paper was to get down the theoretical background in expectations in order that follow up work can be focused on real applications without having to reiterate too much background and algorithmic material.

We acknowledge the paper is long. As the reviewer notes we wanted to be complete in our presentation in order that the work can serve as the foundation for the algorithm, and to make it easier for someone new to the practicalities of $O_2$ A-band remote sensing (as the first author was when he began this work) to pick up the historical background, theoretical basis, and key limitations. As a result we prefer not to split materials into an appendix or supplement as these can be jarring for the reader who does want to go into those details. We also feel that these materials will be useful for when we do show application to real satellite measurements – to test, using those specific examples the Reviewer mentions (Figure 10, Table 4), how histograms of the cost function from real data are, and to get an idea of the uncertainty of real retrievals (and the uncertainty estimates) compared to how well the simulations suggest they "should" do. So we feel that having this documented here is an important part for future works calling back to this study.

**Another point, maybe not a necessary revision, is the idea that a few lines could be added in the conclusion about the overall differences and similarities obtained between OCI and OLCI, as a certain number of results are not shown. It would reinforce the clarity of exploiting OLCI as a proxy to future OCI measurements.**

We have added a few sentences in the conclusion on this.

**Minor comments (not sorted by importance) and points that need clarity :**

**\* about the ratio of reflectance inside/outside the A band**

**Information about Cloud Top Height comes mainly from the ratio of reflectance inside/outside the A band. As the authors are pedagogical in this paper, I wonder if it wouldn't have been possible to add to Figure 2 the dependence of such a ratio to CTH, COT, albedo and fractional depth (FD), that would have shown that the signal is sensitive to CTH, and FD, but less strongly to COT ? A naive question arises : what if the measurement vector consists in ratio of reflectances instead of reflectances? Would have it make a difference in terms of algorithm performance ?**

We experimented with a few additions to Figure 2 but in the end there wasn't a clear way to show the ratio concept as well, without having to make it an additional figure for clarity. So, for length reasons, we decided not to. We did, however, add a sentence talking about the ratio concept and a reference in the text discussing this Figure.

On the topic of expressing the measurement vector as ratios rather than reflectances: provided the window band was expressed as reflectance (as this is the prime driver for COT sensitivity), in theory it should not matter if the others were expressed as reflectance ratios so long as the covariance matrix is formed appropriately. We kept it all as reflectances as it is easier to define the covariance matrix this way (transforming to ratios could make parameterizing elements more difficult if one ends up getting ratios of small numbers in some cases). Where it may make a difference is if some parts of the uncertainty budget are different from what is assumed by the retrieval – in which case ratios might make things better or worse, dependent on whether the the erroneous assumption is masked or magnified by taking ratios (e.g. if one assumed there was no systematic calibration error but in reality there was, then taking ratios might cancel out the effects of this bad assumption). It is the sort of thing that might become clearer once we have real OCI data.

**\* line 243 to 246 :**

**"unfortunately" is maybe not necessary,**

**"sometimes with minimal vertical extent": can you be more precise ?**

We prefer to keep the word "unfortunately" here because in the full context of the sentence we are saying it would be preferable if either (a) fractional geometric depth were invariant or (b) this was readily retrievable from A-band measurements at this spectral resolution, as then uncertainty sources would be smaller.

On the latter point, we have clarified that "minimal vertical extent" means that either a layer without geometric thickness, or tens of m, is assumed by some algorithms.

**\* line 267: error on the writing of "crystals"**

Thank you; this has been corrected in the revised version.

**\* line 279 to 282:**

**the choice you make for ice clouds is a linear profile: but is it a lower triangular profile, or a trapezoidal shape? Also, the ice cloud's vertical profiles in Feofilov (2015) are not all linear and exhibits often a maximum of IWC somewhere between the cloud base and top heights. Actually, the normalized vertical profile of ice clouds depends on the Ice Water Path, see for example Carbajal et al (2013, http://dx.doi.org/10.1063/1.4804794). As the authors point out later, the results are sensitive to the vertical extent of clouds, and also with a magnitude that could be studied, the normalized vertical profile, so that a perspective in follow-up work could be the definition of correct correlation between vertical extent, profile and cloud density (from more statistics that would to go beyond the use of three ARM site measurements). It is important as authors are interested by the forward model uncertainty.**

Thank you for these comments and the reference. We assume a lower triangular profile and in the revised version the text mentions this explicitly. We agree that these idealized profiles are oversimplifications and have added text, and the Carbajal reference provided, mentioning this again. We will likely revisit the ice profile assumptions (and uncertainty budget modeling) at least once after PACE launch and the reviewer's suggestion is exactly the sort of thing to focus on there.

**\* line 403: the authors perturb forward model parameters 'within reasonable ranges'. A question arises about the probability density function they define for each cloud parameters: for example in Table 2 : are the uniform distributions for CTH and for FGD realistic? Do they come from some climatologies? Could have they be better chosen? One can think that, if not optimized, they could eventually degrade the performance of the retrievals.**

The specific distributional choice for FGD arose because we did not have a clear better choice – linking with the above comment and response, this will be revisited post-launch. We've added some text emphasizing this. For CTH we used a uniform distribution in order to obtain sufficient sampling across the range of this retrieved parameter (although real world CTH distributions are not necessarily uniform for a given phase).

**\* line 453: why a systematic error source? Not clear. Please elaborate.**

This is because the plane-parallel assumption tends to lead to errors that are systematic in that, for a given geometry, they are the same sign. We have reworded the text to clarify this.

**\* line 520: you can be maybe more informative in terms of SWIR use.**

We have added wording indicating we're referring to the different spectral gradients of liquid droplet vs ice crystal absorption at swIR bands.

**\* line 545: 'COT clouds are shallower' : what does it mean?**

We think the reviewer is connecting two unconnected sentences here as in Discussions format these are unfortunately split across several pages by the locations of Figures. There are two lines on page 25 where the "COT clouds" sentence ends and the next one, which ends "are shallower", begins. This layout split should not be the case in final journal format.

**\* line 546-550: the statements seem contradictory or the conclusion are not clear: what would be the strategy to account for within-cloud scattering?**

To add extra OCI channels e.g. the $O_2$ B-band. The retrieval does model in-cloud scattering, the limitation is that significant assumptions about cloud vertical structure are necessary when using only the A-band at these spectral resolutions. We have reworded to clarify this.

**\* line 567: precise here how is evaluated the 'retrieval's estimate of retrieval uncertainty'.**

That is the topic of the next line of the paragraph. We've added a pointer back to Equation 4 here in the revised version, for clarity.

**\* line 571-573: comments about Figure 15a and b are not sufficient: why this underconfidence? What have to be revisited?**

We don't know for sure what the reason is. We've expanded the text a little here.

**\* line 583: 'will be useful to data users': please elaborate.**

We feel line 584 covers this as-is, as these applications (data assimilation or use as priors) require well-specified uncertainty models.

**\* line 588: the reference of Desmons et al (2017, JAMC, DOI:10.1175/JAMC-D-16-0159.1) could be added.**

Thank you – we have added this reference in the revised version.

**\* line 610: or in the legend of Figure 17, for clarity, give the definition:**

**(CTPlow – CTP) / (CTPlow-CTPhigh)**

We've added the definition into the text at this point, as suggested.